# Structural dynamics in the water and proton channels of photosystem II during the $S_2$ to $S_3$ transition

Rana Hussein[1,8], Mohamed Ibrahim[1,8], Asmit Bhowmick[2,8], Philipp S. Simon[2,8], Ruchira Chatterjee[2,8], Louise Lassalle[2], Margaret Doyle[2], Isabel Bogacz[2], In-Sik Kim[2], Mun Hon Cheah[3], Sheraz Gul[2], Casper de Lichtenberg[3,4], Petko Chernev[3], Cindy C. Pham[2], Iris D. Young[2], Sergio Carbajo[5], Franklin D. Fuller[5], Roberto Alonso-Mori[5], Alex Batyuk[5], Kyle D. Sutherlin[2], Aaron S. Brewster[2], Robert Bolotovsky[2], Derek Mendez[2], James M. Holton[2], Nigel W. Moriarty[2], Paul D. Adams[2,6], Uwe Bergmann[7], Nicholas K. Sauter[2], Holger Dobbek[1], Johannes Messinger[3,4✉], Athina Zouni[1✉], Jan Kern[2✉], Vittal K. Yachandra[2✉] & Junko Yano[2✉]

Light-driven oxidation of water to molecular oxygen is catalyzed by the oxygen-evolving complex (OEC) in Photosystem II (PS II). This multi-electron, multi-proton catalysis requires the transport of two water molecules to and four protons from the OEC. A high-resolution 1.89 Å structure obtained by averaging all the S states and refining the data of various time points during the $S_2$ to $S_3$ transition has provided better visualization of the potential pathways for substrate water insertion and proton release. Our results indicate that the O1 channel is the likely water intake pathway, and the Cl1 channel is the likely proton release pathway based on the structural rearrangements of water molecules and amino acid side chains along these channels. In particular in the Cl1 channel, we suggest that residue D1-E65 serves as a gate for proton transport by minimizing the back reaction. The results show that the water oxidation reaction at the OEC is well coordinated with the amino acid side chains and the H-bonding network over the entire length of the channels, which is essential in shuttling substrate waters and protons.

[1]Institut für Biologie, Humboldt-Universität zu Berlin, 10115 Berlin, Germany. [2]Molecular Biophysics and Integrated Bioimaging Division, Lawrence Berkeley National Laboratory, Berkeley, CA 94720, USA. [3]Department of Chemistry - Ångström, Molecular Biomimetics, Uppsala University, SE 75120 Uppsala, Sweden. [4]Department of Chemistry, Umeå University, SE 90187 Umeå, Sweden. [5]Linac Coherent Light Source, SLAC National Accelerator Laboratory, Menlo Park, CA 94025, USA. [6]Department of Bioengineering, University of California, Berkeley, CA 94720, USA. [7]Department of Physics, University of Wisconsin–Madison, Madison, WI 53706, USA. [8]These authors contributed equally: Rana Hussein, Mohamed Ibrahim, Asmit Bhowmick, Philipp S. Simon, Ruchira Chatterjee. ✉email: johannes.messinger@kemi.uu.se; athina.zouni@hu-berlin.de; jfkern@lbl.gov; vkyachandra@lbl.gov; jyano@lbl.gov

Water is a necessary ingredient for life on Earth. It is a solvent for all enzymatic reactions and essential for protein folding and activity[1]. In the case of Photosystem II (PS II), which catalyzes the photosynthetic water oxidation reaction in nature, water is also the substrate. The oxidation of water produces most of the $O_2$ in the atmosphere and shapes the biosphere by facilitating the large-scale production of biomass and energy-rich carbohydrates[2]. PS II is embedded in the thylakoid membrane of cyanobacteria, algae, and plants that oxidizes water to dioxygen using light as follows[3] (Eq. 1):

$$2H_2O \xrightarrow{light} O_2 + 4H^+_{lumen} + 4e^-  \qquad (1)$$

The protons ($H^+$), which result from water oxidation, are released into the lumen. PS II carries out this reaction by coupling the one-electron photochemistry occurring at the reaction center with the four-electron oxidation of water at the oxygen-evolving complex (OEC) (Fig. 1a)[4,5]. The OEC consists of a heteronuclear $Mn_4CaO_5$ cluster, which cycles through five intermediate S-states ($S_0$ to $S_4$) that correspond to the abstraction of four successive electrons from the OEC via a redox-active tyrosine residue (Yz)[6]. Once four oxidizing equivalents accumulate at the OEC (metastable or transient $S_4$-state), the release of $O_2$ and the formation of the $S_0$-state take place spontaneously.

During one cycle of the catalytic reaction, the OEC consumes two water molecules; one is introduced into the cycle during the $S_2 \rightarrow S_3$ transition and the second during the $S_3 \rightarrow S_0$ transition (Fig. 1a)[7–12]. In addition to four electrons, four protons are released from the catalytic reaction in the pattern of 1:0:1:2 for S-state transitions, $S_0 \rightarrow S_1 \rightarrow S_2 \rightarrow S_3 \rightarrow S_0$, respectively[8,13–17] (Fig. 1a). The spatially controlled transport of substrate (water) and products (protons and dioxygen) between the catalytic center and the lumenal side of the membrane is essential for efficient catalysis, especially for a multielectron process like the water oxidation reaction. Therefore, it is crucial to understand the role of water and proton channels and the hydrogen-bond network(s) during the reaction process. Several possible water/oxygen/proton channels within PS II have been proposed from computational studies based on structural information obtained at cryogenic temperature at an intermediate resolution[18–22], and more recently based on higher resolution data (Fig. 1a, b)[23–28].

As the $Mn_4CaO_5$ cluster is embedded inside the protein close to the lumenal side of the membrane, it was postulated earlier that water channels and proton exit pathways most likely exist within the complex to ensure proper substrate supply and removal of reaction products (protons and oxygen). Initial work was performed based on the search for cavities and channels using the lower resolution crystal structures, which did not resolve the positions of the waters in the model[18,19,21,22]. A summary of these studies is given in the supplementary material (Supplementary Table 1).

Applying molecular dynamics (MD) simulations gave new insights into, e.g., identifying new channels, characterizing water permeation energetics[18,28], and investigating water diffusion from the bulk[27]. These channels match with some paths identified in the earlier crystal structures (Supplementary Table 1). Among those, Vassiliev et al. demonstrated the first steered MD simulations of solvated PS II[25,28]. This approach, which involves accelerating water permeation by continuous water injection near the OEC, revealed new channels and identified the amino acid residues that narrow the channels and form the bottlenecks and determined the corresponding activation energies for the opening of these bottlenecks. This study also provided two crucial insights concerning water movement. First, water molecules cannot directly permeate from one side of the OEC to the other (Fig. 1b). However, this finding did not exclude the possibility for water to

migrate from one binding site at the OEC to another. Second, none of the channels permit unrestricted access of water to the OEC. This illustrates the difficulty of identifying channels in PS II with static methods (for example, using standard software packages like CAVER or MOLE to map cavities and channels). The potential water channels currently proposed from a series of studies are the O1 channel, the Cl1 channel, and the O4 channel (Fig. 1a, b), and their corresponding names in other studies are summarized in Supplementary Table 1. The O1 channel, aka the large channel[18] starts from near the Ca in the active site, and reaches the lumen at the interface of subunits D1, CP43, and PsbV for branch A and PsbU, PsbV, D2, and CP47 for branch B. Despite the absence of PsbU and PsbV in the PS II of higher plants, the O1 channels are found to be conserved (Supplementary Fig. 1)[27]. The Cl1 channel starts from the Mn4 and connects to the lumen through D1, D2, and PsbO domains for branch A(short) and D2 and PsbO for branch B (long). The O4 channel, aka the narrow channel[18], starts at the O4 side toward the lumen via the D1, D2, and CP43 subunits, before extending to PsbO and PsbU. Similar to the O1 channel, the O4 channel is also conserved in plant PS II[27]. The bottlenecks along each channel that may gate the entrance of the water molecules are shown in Fig. 1b.

Recent advances in X-ray free-electron laser (XFEL)-based room temperature (RT) crystallography enabled us to study the dynamics of the structure of the water network under functional conditions[10,12,29,30]. The ability to take snapshots of the structure at the various time points at RT during the reaction allows for the investigation of water movements and changes in hydrogen-bonding networks in proteins[12]. These studies can provide new insights into the reaction mechanism in PS II by potentially identifying water and proton pathways. They also provide starting models for MD simulations, using RT structures that are the catalytically relevant and functional states, along a reaction trajectory.

In an earlier study by Ibrahim et al.[11], we showed the RT structural changes of PS II together with the kinetics of the Mn oxidation in the OEC during the $S_2$ to $S_3$ transition (at time points 50 μs, 150 μs, 250 μs, 400 μs, and 200 ms after the 2nd flash) under functional conditions, and discussed the major sequence of structural events. We also reported that the OEC remains in the open-cubane configuration, seen in the $S_1$ and $S_2$ states, throughout the $S_2 \rightarrow S_3$ transition[11,12]. There was no indication of the suggested rearrangement of the cluster between an "open-cubane" and a "closed-cubane" structure[31].

In the present study, we focus on the question of mobility of the waters surrounding the OEC. We do that by combing the large dataset we have previously acquired throughout the Kok cycle to obtain a high-resolution structure at 1.89 Å. Regions with more mobility will show a more disordered electron density, whereas regions of less mobility will be more distinct. With this approach, we also identified more waters than previously within the channels described above. These waters were also present in the difference density maps (Fo-Fc) of the individual datasets, but at low sigma level ≤3.0 σ, and hence they were not initially modeled. The presence of these waters in the high-resolution data enabled us to model them into the structure. Introducing these waters into the models for the $S_2$ to $S_3$ time point data and re-refining them led to improved electron density maps and the identification of additional waters.

The $S_2$ to $S_3$ transition is a critical step as it is coupled with the first water binding to the $Mn_4CaO_5$ cluster and the release of one proton. In light of the newly provided information, we investigated the changes in the positions of the amino acid sidechains and the water network(s) that lead to the insertion of water into the open coordination site of Mn (Mn1)[10–12], and the release of protons to identify the possible substrate intake and proton

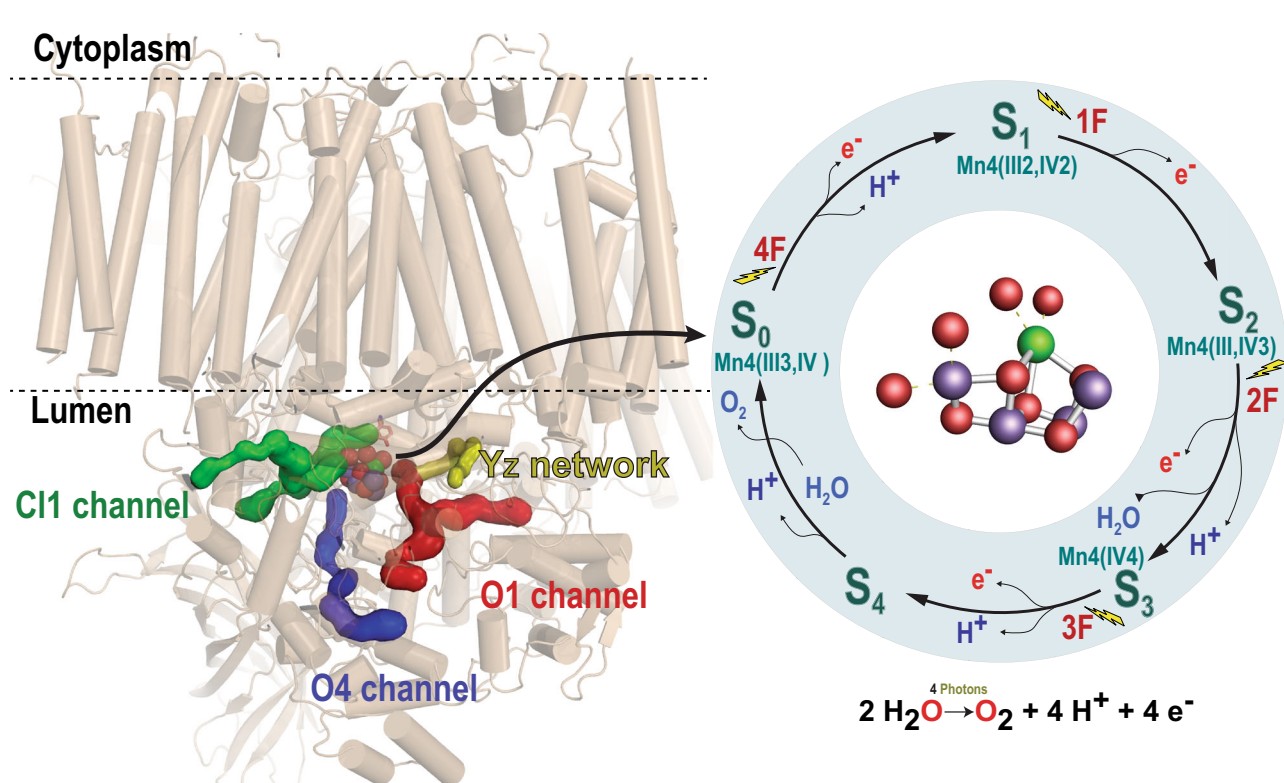

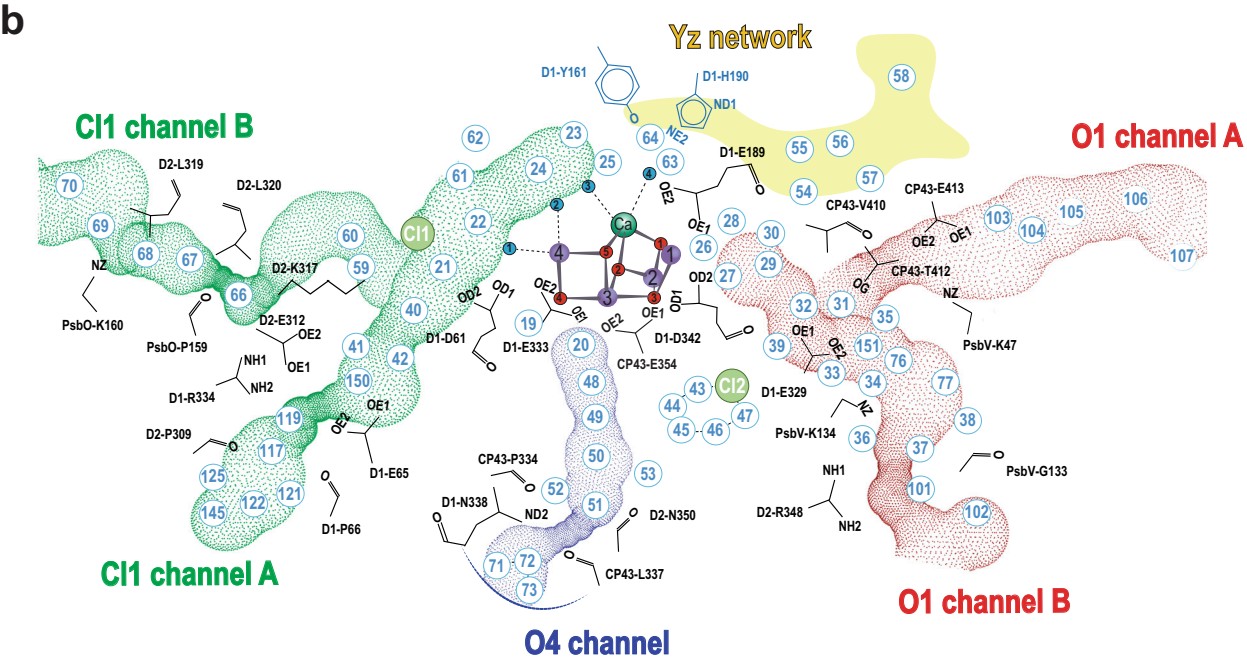

**Fig. 1 An overview of Photosystem II and the main water channels and networks from the OEC to the lumenal side. a** (Left) The structure of PS II showing the membrane-embedded helices and the extrinsic subunits in beige. The OEC and the water channels, in addition to the Yz network, are shown in color. The Kok cycle of the water oxidation reaction that is triggered by the absorption of photons is shown on the right and highlighted with a blue circle. **b** A detailed view of the water channels showing the waters within each channel (O1 Channel red dotted, O4 Channel blue dotted, and Cl1 Channel green dotted). The region highlighted with solid yellow represents the Yz network. Residues involved in forming bottlenecks in the channels are shown in black.

release pathway(s) in relation to previous computational and spectroscopic studies[8,9,32–40]. We also investigated the structure of the water channels and the hydrogen-bonding network through a comparison of the cryogenic and RT crystallographic studies obtained using an XFEL.

## Results

**Mobility of water in the channels during the $S_2 \to S_3$ transition using high-resolution structural data.** A high-resolution dataset was generated by merging more than one hundred thousand high-quality diffraction images collected at RT from PS II crystals in various illumination states and a structure was refined to a resolution of 1.89 Å (Supplementary Table 2). In this high-resolution structure, the waters in the O1 channel have higher B-factor values than waters within the O4 and Cl1 channels (Fig. 2a and Supplementary Fig. 2, see Supplementary Table 3 for water numbering). The average B-factor values for the waters up to approximately 15 Å from the OEC in the O1 channel, O4 channel, and the Cl1 channel are around 38, 31, and 27 Å² respectively. Note that B-factors, or atomic displacement parameters, are directly proportional to the mean square displacement of atoms around their equilibrium position[41]. The waters with a high B-factor in the O1 channel are distributed through the entire channel, starting from the bulk waters (lumen side) to the waters close to the OEC. On the other hand, the waters within the Cl1 have high B-factor values only near the bulk at the lumen side (~33 Å² on average) (Supplementary Fig. 2). The different B-factor values in the channels could be due to the crystal contacts. To check for such potential effects, we compared the B-factors of waters in both monomers, with fixed occupancy during the refinement, as they have different crystal contacts. We show in Supplementary Fig. 3 that in both monomers, the waters in the O1 channel have higher B-factors than those in the Cl1 or O4 channel. See also Supplementary Table 4 and 5 for omit density peak heights and B-factors of waters in the individual datasets.

Furthermore, several Fo-Fc (difference density map) peaks in the high-resolution dataset likely imply partial occupancy of highly mobile waters in the channels. Therefore, the Fo-Fc peaks (≥+3σ) were mapped within all the proposed channels (Fig. 2a). In the O1 channel, the Fo-Fc peaks are distributed through the entire channel, starting from the bulk waters (lumen side) to the waters close to O1 of the OEC. By contrast, the Fo-Fc peaks in the Cl1 and O4 channels appear only near the bulk water on the lumen side.

Besides analyzing the high-resolution dataset, we investigated the change in the normalized B-factor (see SI Methods) of each water within the channels in the dark-adapted state $S_1$, and illuminated states $S_2$, $S_3$, and the four transient time points (50, 150, 250, and 400 μs after the second flash) (Supplementary Fig. 4). The deviation of water positions from the $S_2$-state was also investigated in these time point data (Supplementary Fig. 5).

**Differences in the water network in the cryo and RT structures.** Analyzing the water molecules within the potential channels in the combined high-resolution RT structure and the high-resolution cryo (< 2.0 Å) structures[42,43], shows that the numbers of the waters detected in the channels are comparable at cryo and RT conditions (Fig. 2b and Supplementary Table 6). Our analysis showed that only near the bulk region (up to ~5 Å from the lumen), more waters are observed in the cryo structures, and the most significant difference is observed in the O1 channel B (Fig. 2b).

However, the water networks are different in several locations between the RT and cryo structures. Small non-native molecules,

i.e., glycerol or dimethyl sulfoxide (DMSO), are observed in several crystal structures, Supplementary Table 7, in both the A and B branches of the O1 channel (Fig. 2e, d). These molecules are used as a cryoprotectant[21,42–44], or as an additive during crystallization[10,45]. This implies that small molecules like cryoprotectants can pass through the cavity of the O1 channel. The presence of these molecules in the channel leads to changes in the water network due to their different hydrogen-bonding geometry. For example, a glycerol molecule disturbs the water positions, W38, W76, and W77 of the branch B in the O1 channel (See Fig. 2e).

Unlike the O1 channel, the waters in the much narrower O4 and Cl1 channels are less mobile, as discussed above and also shown by Ibrahim et al.[11]. A small difference in the cryo and RT structures is observed in the O4 channel, where four waters (W50-53) are connected with charged residues and located right before the bottleneck formed by the residues D1-N338, D2-N350, and CP43-P334, -L334 (Fig. 1b). In our RT data, throughout the different illumination states, W50 is connected to three waters (W51-53), creating a "star shape" at the end of the branch (Figs. 1b, and 2c and Supplementary Fig. 6), but it is different from what was observed at cryo temperature (Fig. 2c and Supplementary Fig. 6)[10,42,44]. It was found that one extra water, W603 (PDB ID: 6JLJ)[44] or W757 (PDB ID: 4UB6)[42], is only detected under cryogenic conditions, affecting the hydrogen-bonding water network around W50 (Supplementary Fig. 6).

Differences in the dark $S_1$-state between the cryo and RT structures are also observed in the hydrogen-bonding network around the redox-active Yz (D1-Y161) that mediates electron transfer between the $Mn_4CaO_5$ cluster and the primary electron donor $P_{D1}^+$ (Fig. 2f and Supplementary Fig. 7). At cryogenic temperature, a water (W501 (PDB ID: 6JLJ)[44]/W1117 (PDB ID: 3WU2)[24]) was located close to D1-N298 (Fig. 2f and Supplementary Fig. 7) but it is absent at RT. Hence, D1-N298 is the only connection between the penta-cluster waters W26-27-28-29-30 via W29 and a chain of waters (W54-55-56) on the other side of D1-N298, which in turn are connected to a hydrogen-bonding network to the lumen.

**Structural changes of the waters and sidechains within the channels.** In the next step, we evaluate the motion of waters and amino acid residues along the channels using the RT crystallography data collected for the $S_2$ (200 ms after one flash (1 F)) and the $S_3$ state (200 ms after the 2nd flash (2 F)), and four-time points (50, 150, 250, and 400 μs after the 2nd flash) during the $S_2$ to $S_3$ transition (Supplementary Table 2), where both a proton release and water insertion occur. Extra waters, which are well identified in the combined high-resolution data and also present in the difference density maps (Fo-Fc) of the individual datasets, are included in the models of these time points. Structural refinement, allowing water occupancy changes, results in improving the electron density maps and identifying new features. Figures 3, 4, and 5 show the structural changes in the channels O1, O4, and Cl1 at various time points during the $S_2$ to $S_3$ transition.

**O1 channel.** The mobility of waters in the O1 channel is higher than those in the Cl1 or O4 channels as shown above (Fig. 2a). Prior to the insertion of Ox, the D1-E189 sidechain, which is ligated to Mn1 and Ca in the $S_2$-state, moves away from Ca in the $S_3$-state[12], and Ox becomes a new bridging oxygen between Ca and Mn1. Ibrahim et al.[11] reported that this movement starts within 50μs after the 2nd flash, and by 150μs, the D1-E189 residue is no longer ligated to Ca. This sidechain motion at

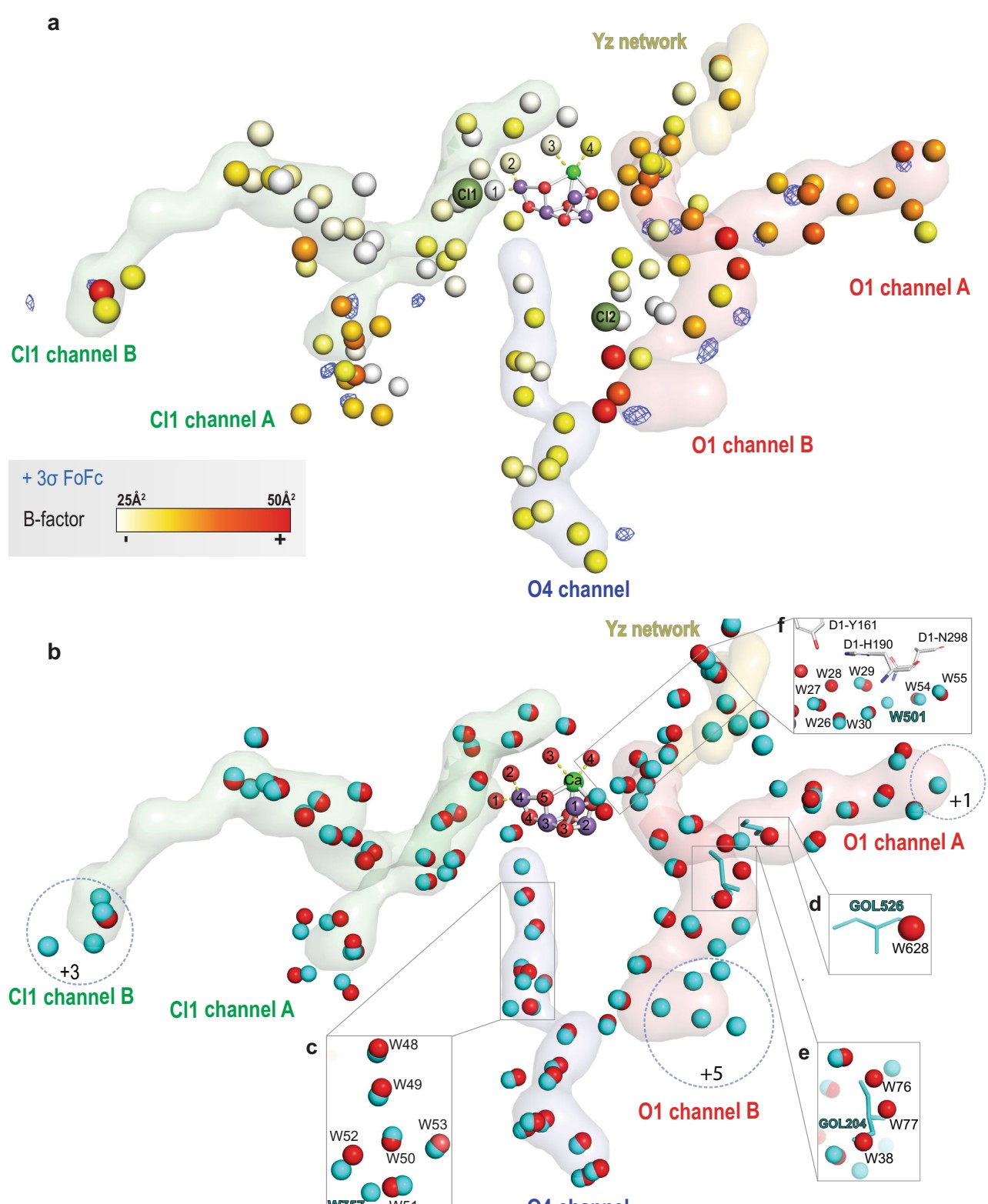

**Fig. 2 Water channels in the 1.89 Å resolution RT structure. a** Fo-Fc, electron density omit map, contoured at ≥ +3σ in blue for all the proposed water channels. The water molecules are represented using a color gradient scale, representing the B-factor of each water (white color for B-factor 25 to red color for B-factor 50). The channels and the network, O1 Channel in red, O4 Channel in blue, Cl1 Channel in green, and Yz network in yellow. **b** Comparison between water channels in the RT and cryo structures. The water molecules detected within 3.5 Å from the water channels in red for the 1.89 Å RT structure (PDB ID: 7RF1) and cyan for the 1.95 Å cryo structure (PDB ID: 4UB6). Extra water molecules in the 1.95 Å cryo structure (PDB ID: 4UB6) detected within 5 Å away from the bulk are highlighted in blue circles. A comparison of the structural differences present in the O4 channel, O1 channel A and O1 channel B between the 1.89 Å RT structure (colored in red and labeled in black) and the 1.95 Å cryo structure (PDB ID: 4UB6) (colored in cyan and labeled in cyan) are shown in (**c**), (**d**), and (**e**), respectively. **f** Structural differences in waters of the Yz network between 1.89 Å RT structure (colored in red and labeled in black) and the cryo structure with PDB ID: 6JLJ (colored and labeled in cyan).

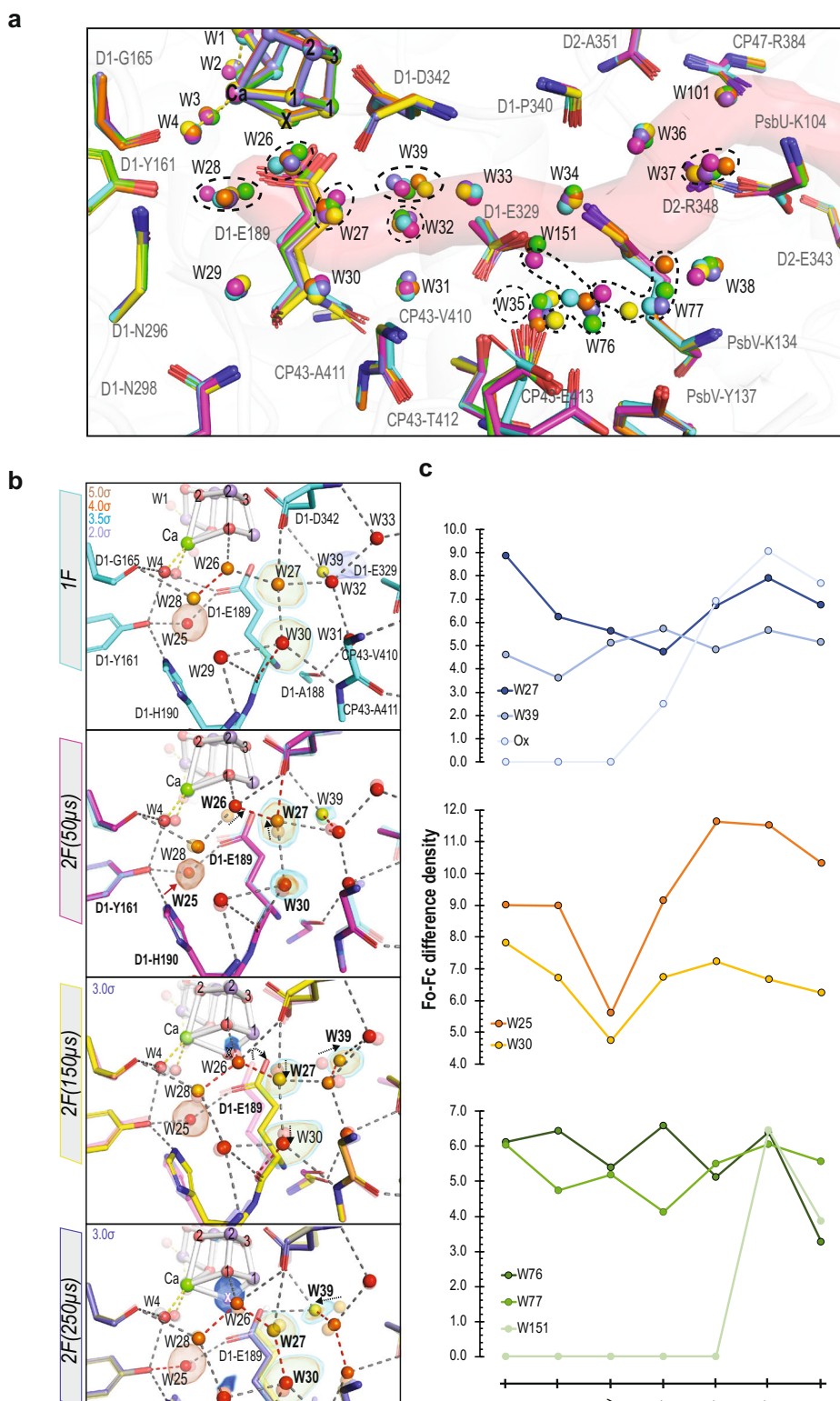

2 F(50μs) is accompanied by a drop of electron density of two waters, W25 and W30, that are hydrogen-bonded to D1-E189 (Fig. 3b, c). W25 is additionally H-bonded to Yz and W3.

Among the waters in the O1 channel, the position of waters that are in close proximity to the OEC (W26, W27, W28, and W39), change the most during the $S_2$ to $S_3$ transition (Fig. 3a). A

significant decrease in electron density and occupancy of W27 were observed at 2 F(150μs) and the W39 electron density increases to reach its maximum at this time point (Fig. 3b, c). This decrease and increase of the electron density at W27 and W39 coincide with the starting of the Ox density build-up at the open coordination site of Mn1. Among the four ligand waters

**Fig. 3 Structural changes in the O1 channel during the S₂ → S₃ transition. a** The structural changes in the O1 channel are shown for all time points in the $S_2 \rightarrow S_3$ transition (1F (teal, PDB: 7RF3) and 2F time points (50 µs: magenta, PDB: 7RF4; 150 µs: yellow, PDB: 7RF5; 250 µs: slate, PDB: 7RF6; 400 µs: orange, PDB: 7RF7; 200 ms: green, PDB: 7RF8)). O1 channel is in red. Waters that show significant movement during the $S_2 \rightarrow S_3$ transition are marked with a black dashed circle. (**B**): Structural changes at the beginning of the O1 channel during the transition at different time points: (1F (teal) and 2F time points (50 µs: magenta; 150 µs: yellow; 250 µs: slate)). Each model overlaid with the model of the earlier time point, shown in a transparent color. The waters are colored based on their occupancies, represented by a color gradient from white to red as shown at the bottom left. The positions for certain waters are confirmed by Fo-Fc omit maps contoured at different σ levels (3.5σ, 4σ) with the exception of W30 at 1F and Ox omit maps contoured at 2σ and 3σ, respectively. The H-bond length is color-coded, as described at the bottom left. Movements of W26, W27, W39, and D1-E189 are marked with black dashed arrows. **c** Difference density heights from the Fo-Fc omit maps for selected waters in the O1 channel and Cl1 channel. Source data are provided as a Source Data file.

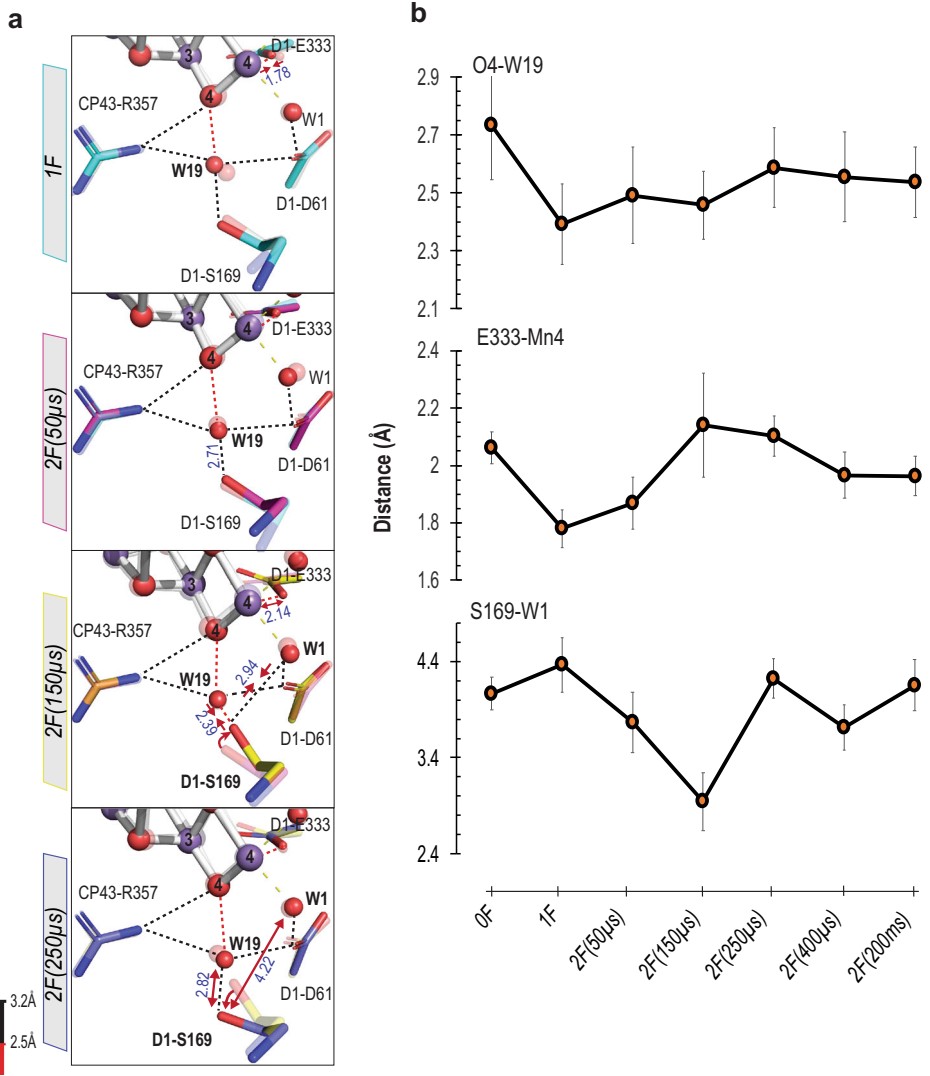

**Fig. 4 Changes near W1 and O4 environment during the S₂ → S₃ transition. a** Structural change in the region of W1 and O4. The structures at different time points are indicated in the left margin, each in solid colors overlaid with the earlier time point, which is in a transparent color scheme (0F (white, PDB: 7RF2), 1F (teal, PDB: 7RF3) and 2F time points (50 µs: magenta, PDB: 7RF4; 150 µs: yellow, PDB: 7RF5; 250 µs: slate, PDB: 7RF6)). The H-bond length is color-coded, as described at the bottom left. Red arrows indicate elongation/shortening of the interatomic distances. The interatomic distances (Å) are shown in blue. 2 F(150 µs) structural data shows rotation in the D1-S169 sidechain affecting the H-bonding network. **b** Distance changes for selected bond lengths within the W1 and O4 environment during the $S_1 \rightarrow S_2$ and $S_2 \rightarrow S_3$ transitions. Error bars represent $+/-$ one standard deviation of each distance calculated by generating 100 randomly perturbed datasets and re-refining, as described in SI Methods and centered around the final refined value for the distance. Source data are provided as a Source Data file.

(W1-4), the B-factor of W4 relative to W1–W3 increases during the S₂ to S₃ transition (Supplementary Fig. 8).

In addition, we observe changes at distal waters, >15 Å from the OEC, in the later time points. For example, W76 and W77, located at ~15 Å from the OEC, along the channel before the

bottleneck formed by residues PsbV-G132, -G133, -K134, D2-R348, show significant movements (Fig. 3a) as seen by the electron density fluctuations (Fig. 3c). At 2 F(400 µs), new electron density appears near the D1-E329 residue close to W34. This density indicates a new water at this position (W151)

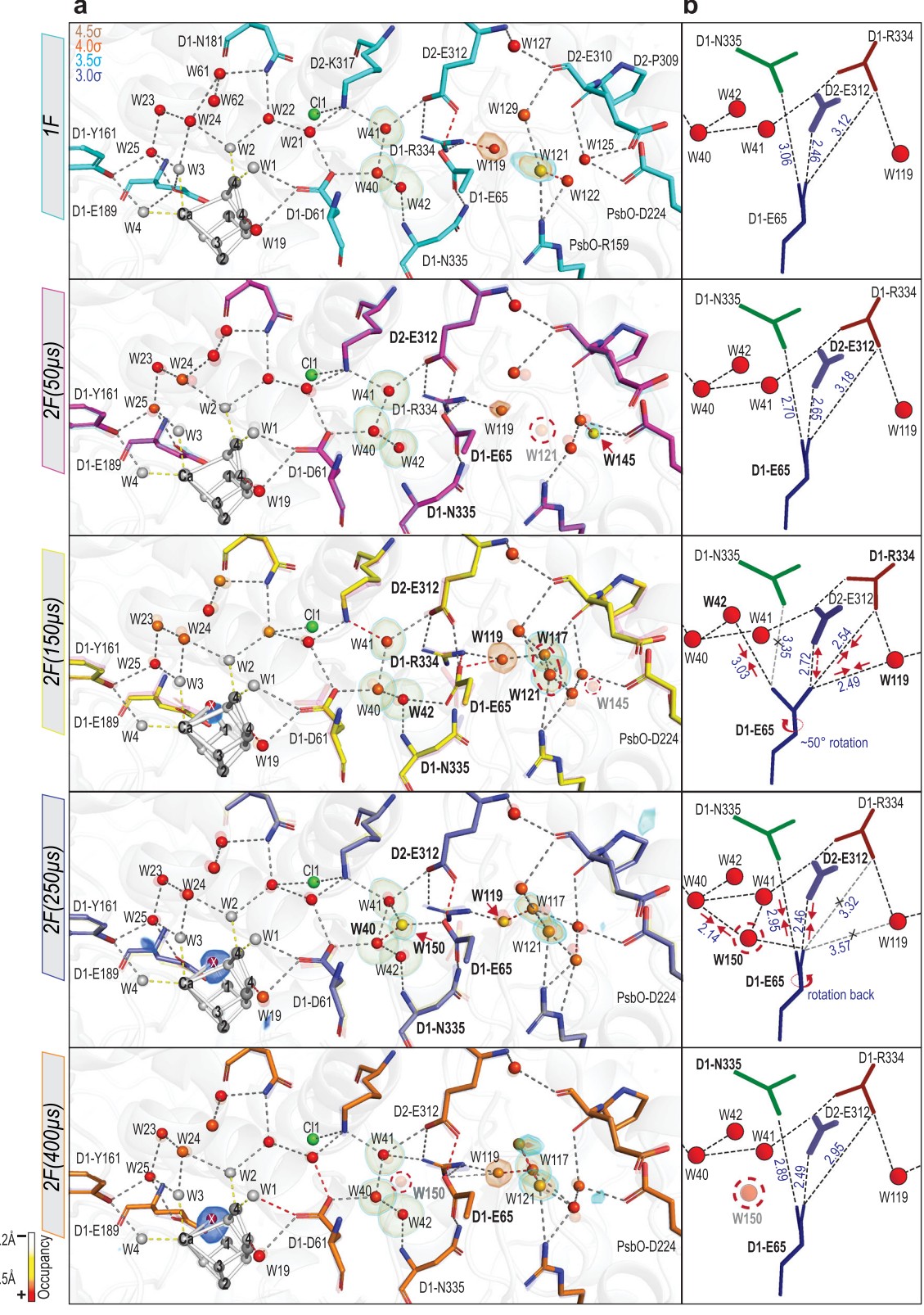

(Fig. 3a, c). In the 2 F(200 ms) data (i.e. $S_3$), its electron density drops significantly.

**O4 channel**. A H-bonded water network in the O4 channel starts from O4 of the OEC and extends through the subunit CP43 until reaching the cavity before the bottleneck formed by D1-N338,

D2-N350, CP43-P334 and CP43-L334 (Fig. 1b). It then extends further to the lumen side through the PsbO and PsbU subunits. The RT structural data reported earlier by Young et al.[30] shows the disappearance of W20, the 2nd water from the OEC in this channel, during the $S_1$ to $S_2$ transition (Supplementary Fig. 9). The same observation was reported by Kern et al.[12], and Suga et al. (W20 named as W699)[44]. This disappearance is due to

**Fig. 5 Structural changes in the Cl1 water channel during the $S_2 \rightarrow S_3$ transition. a** Structural changes in the protein and water molecules during the transition at different time points: (1F (teal, PDB: 7RF3) and 2F time points (50 µs: magenta, PDB: 7RF4; 150 µs: yellow, PDB: 7RF5; 250 µs: slate, PDB: 7RF6; 400 µs: orange, PDB: 7RF7)). Each model overlaid with the model of the earlier time point, shown in a transparent color. The waters are colored based on their occupancies, represented by a color gradient from white to red as shown at the bottom left. The positions for certain waters are confirmed by Fo-Fc omit maps contoured at different σ levels (3.5σ, 4σ) with the exception for W25, W119 and Ox omit maps contoured at (5σ, 4.4σ, and 3σ respectively). The H-bond length is color-coded, as described at the bottom left. Appearance and disappearance of W121, W117, W121, W145, and W150 at different time points are marked with a red dashed circle. **b** Model of the structural changes around D1-E65, D2-E312, and D1-R334. The H-bonds are shown in dashed black line and up to 3.2 Å. Red arrows indicate elongation/ shortening of the interatomic distances. The interatomic distances (Å) are shown in blue. The rotation of the D1-E65 at 2F (150 µs) is marked by a red circle arrow. The appearance and disappearance of W150 at 2F (250 µs) and at 2F (400 µs), respectively, are marked with a red dashed circle.

either W20 moving away from its position in the channel or having an increased mobility after the 1st flash. In both scenarios, the hydrogen-bonding network along the O4 channel becomes disconnected from the OEC in the $S_2$-state and is restored only in the $S_0$-state. In addition, the positional changes of waters W19, W49, and W50 (Supplementary Fig. 9) along this channel, also result in keeping the hydrogen-bonding network disrupted during the $S_2$ to $S_3$ transition. We also observed changes at the beginning of the channel, in the O4-W19 – D1-S169 - W1 network, during the $S_2$ to $S_3$ transition (Fig. 4). As the changes in this area are also related to the changes in the Cl1 channel, we discuss them together in the next section.

**Cl1 Channel**. The Cl1 channel that starts from the Mn4 side of the OEC involves one of the chlorides (Cl1) near the OEC and D2-K317 and D1-D61 residues that form a bottleneck (Fig. 1b). Through D1-D61, the Cl1 channel is connected to W19, which is also the first water in the O4 channel. Along the Cl1 channel, we observed sequential changes during the $S_2$ to $S_3$ transition along branch A around the bottleneck that is formed by D1-E65, D1-P66, D1-V67 and D2-E312, and in the area of O4, W19, D1-S169 and W1 (shown in Figs. 5 and 4, respectively).

After the 1st flash, the Mn4-$O_{D1-E333}$ distance is shortened from 2.06 to 1.78 Å, which is accompanied by the shortening of the O4-W19 distance by 0.34 Å with a standard deviation of 0.14 (2.73 Å→2.39 Å) (Fig. 4a, b). This could be due to the oxidation of Mn4 (from III to IV) after the 1st flash, which likely influences the Mn4-O4 oxo-bridge. By 150 µs after the 2nd flash (2 F(150 µs)), the sidechain of D1-S169 moves significantly (Fig. 4a), decreasing the distance between S169 and W1. Concomitantly, the D1-E65 sidechain, a pivotal contributor to the bottleneck, rotates ~50°, which results in a drastic rearrangement of the H-bond network in this area (Fig. 5a, b). The rotation of D1-E65 disturbs its hydrogen-bonding interaction[46] to D1-N335 (2.70 Å → 3.35 Å). This also weakens the interaction between D1-E65 and D2-E312 (2.65 Å→2.72 Å), while strengthening the interactions to D1-R334 (3.18 Å → 2.54 Å). The rotation of D1-E65 alters the interaction to W119 (3.47 Å → 2.49 Å), and W42 (3.46 Å → 3.03 Å) (Fig. 5). The analysis of the radius of this bottleneck of the Cl1 channel during the $S_2$ to $S_3$ transition showed that the D1-E65 rotation, at 150 µs after the 2nd flash, reduces the radius of the bottleneck (Supplementary Fig. 10). We note that this channel appears too narrow for water transport, but that transient openings occurring, at any studied time point, only in small fractions of the centers cannot be excluded.

By 250 µs into the $S_2$ to $S_3$ transition (2 F(250 µs)), the D1-E65 side chain rotates back to its original position, and concomitantly a new water (W150) with ~55 % occupancy appears in hydrogen-bonding distance to D1-E65 and within close distance to W40 (~2.1 Å). This is accompanied by shortening of the distance between D1-E65 and D2-E312 from 2.72 to 2.46 Å and a decrease in the electron density of W119 (Figs. 4 and 5) and its occupancy

from ~80% at 2 F(150 µs) to ~53% at 2 F(250 µs). However, the omit densities of W119 fluctuates during the $S_2$ to $S_3$ transition (Fig. 5a).

In addition to the side chain motions described above, we observed the appearance/disappearance of some additional waters along the Cl1 channel. In the region after the bottleneck, W117 observed in the $S_1$-state is not present in 1F ($S_2$), but present at 2 F(50 µs) (Fig. 5 and Supplementary Fig. 11). W121 is not observed in the 2 F(50 µs) data, while extra electron density near PsbO-D224 modeled as W145 was observed. In 2 F(150 µs), these changes are reversed, i.e., the W117 and W121 densities reappeared, together with the disappearance of W145.

By 400 µs into the $S_2$ to $S_3$ transition (2F(400 µs)), the Cl1-channel environment is rearranged back to one being similar to the $S_1$-state (Fig. 5), and no significant changes are observed between the $S_1$ and the fully evolved $S_3$-state (2F(200 ms)) (Supplementary Fig. 11).

## Discussion

Our study investigates the motion of waters and surrounding amino acid residues using snapshots of the RT crystal structures of PS II to identify substrate water and proton release channels. We focus on the $S_2$ to $S_3$ transition step, where one electron and one proton are released, and one water molecule comes into the OEC to the open coordination site of Mn1 as a bridging oxo or hydroxo ligand between Mn1 and Ca[10–12]. The water positions are largely preserved between the RT and cryogenic structures in the dark-adapted state, suggesting that most of the waters along the potential channels are highly structured, except at the exit of each channel into the bulk. There are, however, some differences, which likely have important implications for the interpretation of the proton relay (Fig. 2B). We have also observed sequential changes of several water positions at RT along the course of the $S_2$ to $S_3$ transition after the 2nd flash (Supplementary Fig. 5). Based on the above observations, we discuss the likely channels for proton release and water intake.

For substrate intake pathways, the O1, O4 and Cl1 channels have been proposed based on theoretical studies[18,25,27,28,47–49]. Therefore, determining the substrate channel will help reducing possible substrate insertion pathways in the OEC that lead to Ox in the $S_3$ state. The combined 1.89 Å high-resolution structure of PS II in this study helped model partially occupied waters in the crystal structures of the individual timepoint data. Previously these water densities were visible in the timepoint data but were not included in the model due to the uncertainty[11,12]. In addition, the high-resolution data shows several Fo-Fc peaks (Fig. 2a), that are spread along the O1 channel, but not in the Cl1 or O4 channel. This is most likely due to waters occupying different positions at different time points or the presence of new partially occupied waters included in the combined data, implying more mobile waters in the O1 channel (Fig. 2a). Besides, the combined high-resolution structure shows that the waters in the O1 channel have significantly higher B-factors than those in the Cl1 channel

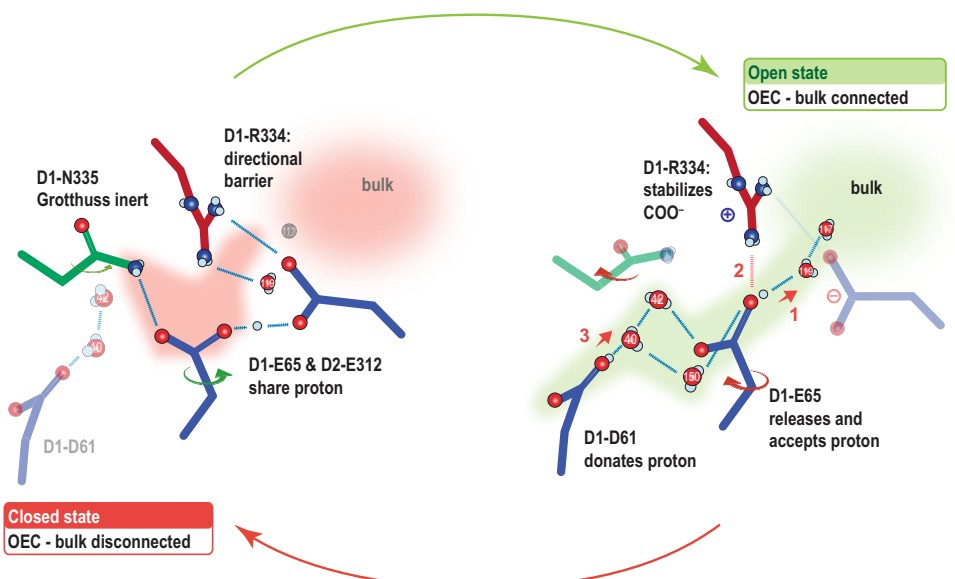

**Fig. 6 The proposed proton gate around D1-E65, D2-E312, and D1-R334 in the open and closed state.** In the closed state, the H-bonding network connecting the OEC to the bulk is disrupted by D1-N335 and D1-R334, while in the open state, the OEC is connected via D1-E65 and waters to the bulk. The opening of the gate by the rotation of D1-E65 could be caused by the protonation of D1-D61 and the subsequent rearrangement of the H-bonding network. We hypothesized that the proton released towards the bulk was shared between the D1-E65 and D2-E312 before. The deprotonated D1-E65 can then be stabilized by approaching D1-R334. Rotation of D1-E65 back to the original position and closing of the gate may be caused by proton transfer from D1-D61 to D1-E65 and the subsequent repulsion and attraction of the newly arrived proton by D1-R334 and D2-E312, respectively.

or O4 channel (Fig. 2 and Supplementary Figs. 2 and 3). Along the same line, deviations in the positions and the high fluctuations in the normalized B-factors of the waters in the O1 channel at the different time points during the $S_2 \rightarrow S_3$ transition confirm the higher mobility of water in the O1 channel in comparison to Cl1 and O4 channels (Supplementary Figs. 4 and 5).

Moreover, in the crystal structures with glycerol or DMSO present in the crystallization process[10,24,42,44], these molecules were visible in the O1 channel, but not in the Cl1 or O4 channel. The FTIR data by Kato et al. reported that the S-state turnover efficiency of crystals (with glycerol) is slightly lower than that of the solution sample (without glycerol) in the $S_2$ to $S_3$ and $S_3$ to $S_0$ transitions, the steps where the substrate water insertion is involved[33,50]. We hypothesize that such a decrease may be caused by the slightly altered water network in the O1 channel in the presence of small additives that influence the access or mobility of waters. We, however, note that the effect of such small additives in the O1 channel is minor and does not block the water oxidation reaction.

The new water near D1-E329 detected in the 2 F(400 μs) data after the completion of the Ox insertion at the Mn1 site may also support the O1 channel being the water intake channel from the lumenal side of the membrane. However, we cannot determine whether branch A, or B, or both serve as a pathway. Regarding the possibility of the Cl1 channel being the water intake pathway, we think this is unlikely due to the following observations: First, the mobility of waters in the channel is lower than that of the O1 channel. Second, no structural changes were observed along branch B. Third, the structural changes observed at the bottleneck region in branch A do not allow a water to pass through. While D1-E65 rotated by ~50°, this flipping motion does not open up the Cl1 channel (Supplementary Fig. 10). We think that the reversible motion of D1-E65 and the presence of less mobile waters makes the Cl1 channel more suitable to serve as a proton pathway (Fig. 6). This hypothesis is also supported by a study of oxidative modifications in PS II by Weisz et al., where the authors identified the O1 channel as one of the possible pathways for

water intake, but not the Cl1 branch A[51]. Nevertheless, branch B of the Cl1 channel was also found to be oxidized by ROS formed at the OEC.

Building upon the hypothesis of the O1 channel being the water intake pathway, we explore the question of how the water in the O1 channel ends at the open coordination site of Mn1 as a bridging ligand, Ox (oxo or hydroxo), to Mn1 and Ca during the $S_2$ to $S_3$ transition[10–12]. Prior to the Ox insertion, our timepoint data showed the changes of the interaction between the redox-active Yz and H190, due to the oxidation of Yz by the primary electron donor $P_{D1}^+$ following charge separation[11]. A detailed description of the structural changes within this region was reported in Ibrahim et al.[11]. The main change is the tilting of the histidine side chain and the backbone, providing the driving force for the D1-E189 to move away from Ca[11]. While the motion of E189, ligated to Mn1 and located close to the O1 channel, changes its position, there is not enough space for the direct insertion of water from the O1 channel to occur.

An alternative route for the Ox insertion is via W3, which is a ligand of Ca[11]. Ugur et al. suggested that W3 moves to the open coordination site of Mn1[52], which was later supported by FTIR[53] and other theoretical studies[37]. If W3 is the entrance of substrate water to the Ox from the O1 channel, it likely needs to come via W4 first, which is refilled from the water wheel-like ring (penta cluster) of W26, 27, 28, 29, and 30 (Fig. 7). In this case, Ca likely plays a pivotal role to shuffle water from W4, W3, and then to Ox. Our data show that the B-factor of W4 relative to W1–W3 is higher during the transition (2 F(50 μs), 2 F(150 μs), and 2 F(400 μs)) (Supplementary Fig. 8). Among the O1 channel waters proximal to the OEC, W28 is found in a suitable geometry to refill W4, and W28 could be refilled by W26/W27 (Fig. 3).

Another possibility is that W25 replaces W3, as W25 is located close to D1-E189 and W3. W25 shows reduced electron density at 50 μs after the 2nd flash (Fig. 3). This drop of the electron density is likely related to the loss or weakening of the H-bond to Yz due to Yz oxidation, thereby weakening the H-bond between W25 and the side chain of E189, possibly priming it for replacing W3.

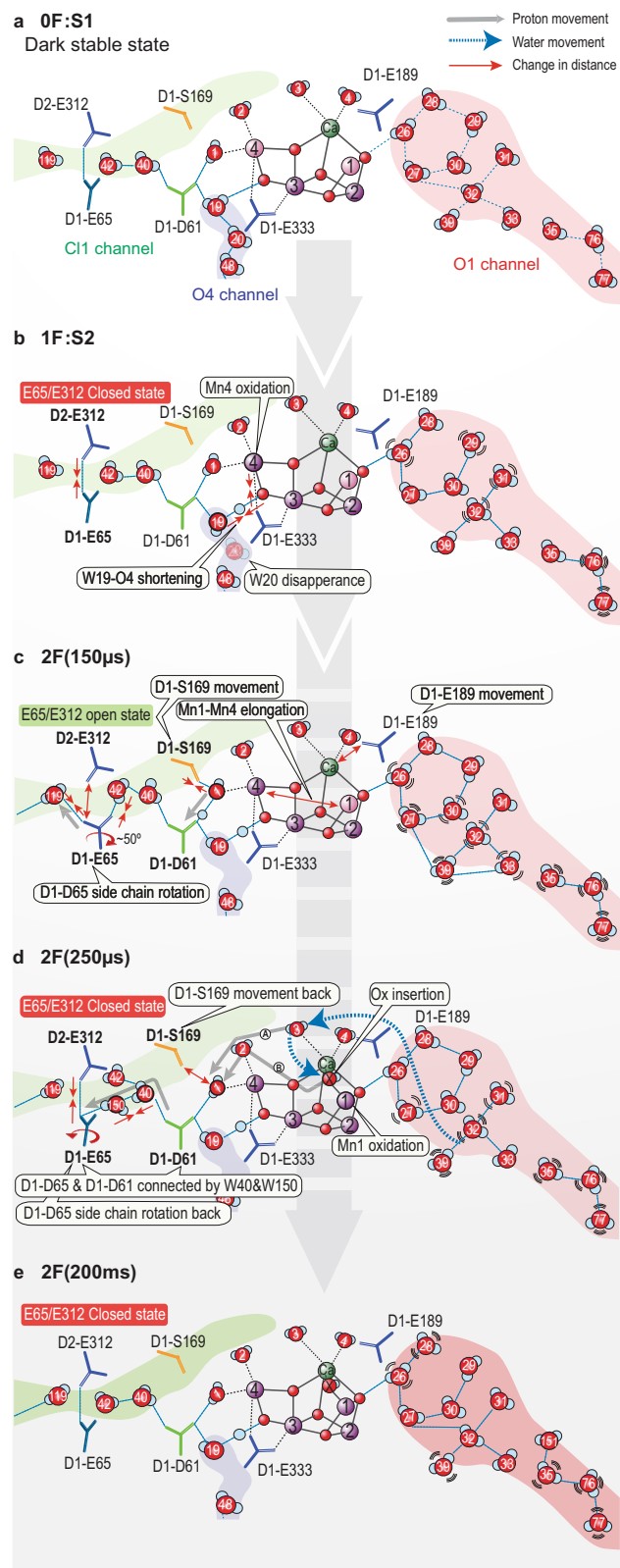

**Fig. 7 Schematic summarizing the structural changes in the O1, Cl1, and O4 channels leading to the first water insertion and proton release during the $S_2 \rightarrow S_3$ transition.** Mn1 and Mn4 oxidations from (III) to (IV) are shown as a color change from pink to purple. The gray arrow represents the possible proton pathway, while the blue dashed arrow represents the potential stepwise water insertion pathway.

Accompanied by electron transfer and the intake of a water, the egress of a proton is also required during the $S_2 \rightarrow S_3$ transition. The Cl1 and O4 channels as well as the Yz network, have been proposed in the literature as a proton release pathway[32,34,39,40,54–56]. The current structural study provides several indications of the Cl1 channel being the proton exit pathway in the $S_2$ to $S_3$ transition. Below we discuss the possibility of proton transfer for all three pathways and provide support for the hypothesis of the Cl1 channel being the proton pathway in the $S_2$ to $S_3$ transition.

Recently the O4 channel was discussed as a possibility for the egress of a proton during the $S_0 \rightarrow S_1$ transition[36]. Theoretical studies suggested that the water chain in the O4 channel provides a downhill proton transfer[39,57]. In the $S_1$ to $S_2$ transition (Fig. 3b), the disappearance of W20 was observed near the OEC at the beginning of the channel near O4[12,30,44]. The dislodging of W20 disconnects the hydrogen-bonding network of the O4 channel from the OEC in the $S_2$-state, and therefore it is unlikely that the O4 channel can serve as a proton release pathway during the $S_2$ to $S_3$ transition. This channel has been proposed as a proton release channel during the $S_0$ to $S_1$ transition[36,39,57,58]. If the proton release during the $S_0 \rightarrow S_1$ transition is via the O4 channel[36], the current observation raises the possibility that the proton release pathway may differ in each S-state transition.

The Yz network, which was postulated to connect Yz to the lumen via D1-N298, has been suggested to be involved in a proton pathway based on the nature of the [YzO⋯H⋯Nε-His] H-bond[24,59,60] and the water positions in the cryogenic structure[32,34]. In the RT structural data, however, one water (W501), which could be essential for proton transfer via this network (Supplementary Fig. 7), is missing. If this water does not exist or is highly mobile, the Yz network would require a proton transfer through an asparagine (D1-N298)[27,34] residue that is not generally considered suitable for proton relay. The Yz network ending at the lumen surface residue PsbV-K129 is also rich in other asparagine residues (D1-N301-303-322), and these require tautomerization or amide rotations to allow proton transferr through them[32]. In our current crystal structures, we do not observe any structural changes of these residues during the $S_2 \rightarrow S_3$ transition (Supplementary Fig. 12). An alternative proton pathway that involves waters would require an interaction between W57 and W58. These two waters, however, are separated by 6.4 Å in the $S_2$-state and do not show any substantial distance changes in all the timepoint models. Therefore, we concluded that the Yz network, based on the RT structural data, is unlikely to be a proton release pathway.

The Cl1 channel has been proposed as a proton release pathway during the $S_2$ to $S_3$ transition in many studies in the literature[26,54,55,61], and our current structural observation prefers this assignment. We observe structural changes around D1-E65, D2-E312, and D1-R334 at 150 μs after the 2nd flash, which are reversed by 250 μs. We hypothesize this motion might be triggered by the excess positive charge after Yz oxidation, and slower protonation and H-bonding rearrangements, and is related to the opening of the channel to proton transfer and release as illustrated in Fig. 6 and described in the following. In the $S_2$-state, D1-E65 and D2-E312 share a proton as indicated by the short H-bonding distance of around 2.5 Å and confirmed by simulations[55]. This suggests that in the $S_2$-state and up to 50 μs after the 2nd flash (2 F(50μs)), the Cl1 channel is in a 'closed state' for proton egress, since D1-E65 is H-bonded to D1-N335, which is inert to a Grotthuss proton hopping mechanism, and D1-R334 is H-bonded to W119, functioning with its positive

charge as a directional barrier (Figs. 5b and 6). At 150 µs after the 2nd flash, however, we observe a new conformation of D1-E65 that rearranges the E65/E312 region and hydrogen bonds to W119 (Figs. 5b and 6). This rearrangement forms an 'open state' for an effective proton transfer to the other side of the bottleneck. The proton transfer from the protonated D1-D61 to a deprotonated D1-E65 or E312 was recently reported to be exothermic via one or two waters[55]. This implies that E65-E312 can accept a proton from the OEC only after releasing a proton towards the bulk (Fig. 6). This conformation is stabilized by a newly formed H-bond between E65 and R334, preventing the released proton to return. A proton could now be transferred from D61 to E65 via at least two possible water chains involving W40/42 or the alternative water position of W150. The newly arrived proton will be repelled by R334 and attracted by E312, so that by 250 µs the D1-E65 side chain rotates back to its original position. At 400 µs, W150 is not present anymore, and with it the last indication of a fraction of open states.

With the current data, however, we cannot conclude when exactly the proton is released to the bulk from the Cl1 channel. Early changes (50 µs after the 2nd flash) are observed around the water network at the PsbO-D2 interface (Fig. 5a), which may indicate a fast, long-range electrostatically triggered proton release of surface carboxylates as found in PS II membrane particles from plants[14,62]. The short H-bonding interaction of E65 and R334 in the open state of the gate at 150 µs may indicate that a proton is already released from the gate in line with a suggested fast refilling of the earlier deprotonated site[62]. Alternatively, the gate may stay open until the proton is released in an apparent single proton hopping event occurring with Mn oxidation[56]. This is supported by the observation that at 150 µs the averaged distance between E65 and E312 is still ~2.7 Å, indicating that in some centers a proton is still shared between both residues and the proton release cannot be terminated, yet. In any case, the gate opening and closing can directly explain the involvement of multiple protonatable side chains as observed in pH-dependent oxygen activity[63] and the reported pH dependency of the proton-coupled electron transfer (PCET) of Mn oxidation in the $S_2$ to $S_3$ transition[56,64]: In the open state (Fig. 6), when the proton is still located at the gate, a low pH at the proximal bulk may prevent it from releasing its proton. This will slow down the deprotonation of D61 and also the deprotonation of the newly inserted substrate water (Ox).

In Fig. 7, we summarize the structural sequence of events during the $S_2$ to $S_3$ transition, based on our current observation of the structural changes starting from the $S_2$ formation; we integrated the proposed proton release and water insertion process with the O1 channel as a water intake, and the Cl1 channel as a proton release pathway. Upon the $S_2$-state formation (Figs. 7a–b), Mn4 is oxidized, which is indicated by the shortening of the Mn4-$O_{E333}$ distance, as reported previously[11]. This redox change may trigger the structural changes around the area; the W19-O4 distance is shortened, likely due to equal sharing of the proton between them, leading to the weakening of the hydrogen-bond between W19 and W20. The W20 density is not visible in the $S_2$-state, only reappearing in the $S_0$-state. As a consequence, the hydrogen-bond network from the OEC to the O4 channel is disconnected during the $S_2$ to $S_3$ and $S_3$ to $S_0$ transitions.

Upon the 2nd flash, Yz oxidation triggers the movement of the Yz-His190-E189 region that shifts D1-E189 away from Ca and the elongation of Mn1-Mn4 is observed by 150 µs after the 2nd flash (Fig. 7c). In the same timescale, D1-S169 rotates to be in H-bond distance to W1 (Figs. 4 and 7c). One plausible explanation for this motion is that W1 releases a proton via D1-D61

and becomes an $OH^-$, that interacts with S169. The low barrier proton release of W1 via D1-D61 was suggested from the DFT calculations for the $S_2$ to $S_3$ transition[65]. Also, the role of D1-E65 and D1-D61 in the proton release has been widely discussed[20,27,35,46,54,55,59,66–69]. The new conformation of D1-E65/E312 with W119 forms an 'open state' for proton release towards the bulk by 150 µs (Fig. 7c). By 250 µs (Fig. 7d), the changes in the E65/E312 and D1-S169-W1 region are reversed, after D1-E65 gets protonated from D1-D61 via W40 and W150/W42, and the proton gate is closed. As indicated by the increased distance of D1-S169 to W1, the $OH^-$ (W1) ligand becomes protonated again. Proton motions most likely correlate with the PCET reaction that involves the Mn1 oxidation and the Ox insertion from the O1 channel. Since the water is easier to be deprotonated when bound to a metal-like Mn or Ca, we propose water is first bound to Ca or Mn prior to being inserted at the open coordination site of Mn1, possibly via W3. The time constant of the Mn1 oxidation we observed by Mn $K\beta_{1,3}$ XES is around 350 µs[11]. Therefore, the reprotonation of W1 can proceed from the newly inserted Ox. By 400 µs, all the movement around the OEC is complete, and no major changes are observed in the channel regions between 2 F(400 µs) and the $S_3$-state 2 F(200 ms), Fig. 7d.

In summary, we investigated the water and proton channels that connect the OEC to the lumenal bulk water. Based on the RT structures, we propose that the O1 channel with mobile waters is suitable for water intake from the bulk to the OEC, while the more rigid network of the Cl1 channel branch A, formed with amino acids and waters extending through W1, D1-D61, to D1-E65, is suitable for proton relay during the $S_2$ to $S_3$ transition. Based on the observed structural changes, we hypothesize that D1-E65, D2-E312, and D1-R334 form a proton gate by minimizing the back reaction, thus regulating proton release from the OEC to the bulk. We also note that different proton release pathways may be used during different S-state transitions. The current study is a first step showing how the coordinated motion of amino acid residues and the water network is key to spatially control substrate and proton transport required for a multi-electron/proton process like the water oxidation reaction.

## Methods

**Sample preparation.** Crystals ranging in size from 20 to 60 µm were obtained from PS II dimers of *T. elongatus*[70,71] and were used for XRD measurements in 0.1 M MES, pH 6.5, 0.1 M ammonium chloride, and 35% (w/v) PEG 5000. PS II crystals are highly active in $O_2$ evolution, show no Mn(II) contamination[72], and turnover parameters and S-state populations under our experimental conditions were determined by membrane inlet mass spectroscopy[73] as described previously[71].

**Analysis of O1 channel in cyanobacteria and higher plants.** The O1 channel was mapped by Caver 3.0 Pymol plugin[74] using the RT crystal structural of cyanobacterial PSII (PDB: 7RF2) and the Cryo-EM structure of plant PSII (PDB: 3JCU). The channels start near the Ca side and extend through the D1 subunit (Supplementary Figure 1). The termini of the channels in cyanobacteria were found to be between subunits CP43 and PsbV, crosssponding to O1 channel A (Supplementary Figure 1a), or between subunits D2, CP47, PsbV, and PsbU, corresponding to O1 channel B (Supplementary Figure 1c). The termini of the channels in the plant were found to be either between subunits CP43 and PsbP (Supplementary Fig. 1b) or between subunits D2, CP47, PsbP, and PsbQ (Supplementary Figure 1d). Our results showed that the cyanobacterial O1 channel A and O1 channel B are structurally conserved in plants as they correspond well with the channel detected in PS II of plants that proceeds along subunits CP43 and PsbP and subunits D2, CP47, PsbP, and PsbQ, respectively.

**X-ray data collection.** The crystallography data were collected at the MFX instrument of LCLS at the SLAC National Accelerator Laboratory, Stanford[75,76]. XRD and XES of PS II crystals were measured using X-ray pulses of ~40 fs length, at 9.5 keV, with pulse energies of 2-4 mJ, and with an X-ray spot size at the sample of ~3 µm in diameter. XRD data were collected using a Rayonix 340 detector, operating in the 3-by-3 binning mode, at a frame rate of 20 Hz. The sample was delivered to the X-ray interaction region using the previously described Drop-on-

Tape setup[77]. Illumination conditions for populating different S-states are described in the following references[12,71].

**X-ray diffraction data processing**. From the data collected at LCLS for different illumination states, as described previously[71], a total of 262,254 integrated lattices were obtained using *psana*[78] to read the images and *dials.stills_process* for integration, with a target unit cell of $a = 117.0$ Å, $b = 221.0$ Å, $c = 309.0$ Å, $\alpha = \beta = \gamma = 90°$, and the space group $P2_12_12_1$. Signal was integrated to the edges of the detector and subsequently, a per-image resolution cutoff was used during the merging step. Integrated intensities were corrected for absorption by the kapton conveyor belt to match the position of the belt and crystals relative to the X-ray beam[77]. Ensemble refinement of the crystal and detector parameters was then performed on the data using *cctbx.xfel.stripe_experiment* and improved the unit cell distribution and final isomorphous difference maps. After ensemble refinement and filtering out 32,428 lattices that belonged to a different crystal isoform, a total of 229,810 integrated lattices was obtained with an average unit cell of $a = 117.0$ Å, $b = 221.7$ Å, $c = 307.6$ Å, $\alpha = \beta = \gamma = 90°$ and the space group $P2_12_12_1$.

Image sets were also culled to include only images extending past 3 Å, as we have done previously for PS II datasets to improve statistics by removing contamination due to low-quality images[12]. The remaining integrated images were merged using *cxi.merge* as described previously[12,71]. A combined dataset at 1.89 Å, containing images from all illumination states was first obtained using a previously obtained reference model with no restrictions on the unit cell parameters. The combined dataset was obtained by merging reflections from all the images in the experiment, which had reflections extending beyond 3 Å. This included all the stable intermediate states of the Kok cycle (S₁, S₂, S₃, and S₀) as well as the timepoints between those intermediates. In total, 111,922 images were merged to yield a combined dataset cut at 1.89 Å by using the criterion of monotonic falloff of the $CC_{1/2}$ and the resolution where the average multiplicity falls below 10. The $R_{merge}$ for the dataset is 11.7% (91.4% in the highest resolution shell, 1.922-1.890 Å). This combined mtz file was used to generate a refined pdb model of the combined dataset as described below, and this pdb model was used as the reference model for merging the separate illumination states. The unit cell outlier rejection option in *cxi.merge* was used to remove images with a unit cell that differed by more than 1% from the reference model, so a prefiltering step was not necessary.

Final merged datasets were acquired for the combined dataset, 0F, 1F, 2F(50 μs), 2F(150 μs), 2F(250 μs), 2F(400 μs), and 2F states to resolutions between 2.27 and 1.89 Å, containing between 4464 and 111,922 images (Supplementary Table 2). Please note that the merged datasets for the individual illumination conditions are the same as those used in ref. 2.

**Model building and map calculation**. About 112,000 diffraction images, collected at room temperature from PS II crystals in various illumination states and falling within a 1% unit cell tolerance cutoff, were merged, which resulted in a high-resolution dataset. Initial structure refinement against this combined dataset at 1.89 Å was carried out starting from a previously acquired high-resolution PS II structure in the same unit cell (PDB ID: 5TIS)[30] using *phenix.refine*[79–81]. The $R_{free}$ set was 0.89% of the total reflections and was created by extending the resolution of the $R_{free}$ set used for the refinement of the individual datasets. As a result, the $R_{free}$ reflections for the combined dataset contain all the $R_{free}$ reflections used for the individual datasets. B-factors were reset to a value of 30 and waters were removed. After an initial rigid body refinement step, *xyz* coordinates and isotropic B-factors were refined for tens of cycles with automatic water-placement enabled. Custom bonding restraints were used for the OEC (with large σ values, to reduce the effect of the strain at the OEC on the coordinate refinement), chlorophyll-*a* (CLA, to allow correct placement of the Mg relative to the plane of the porphyrin ring), and unknown lipid-like ligands (STE). Custom coordination restraints overrode van der Waals repulsion for coordinated chlorophyll Mg atoms, the non-heme iron, and the OEC. Following real-space refinement in *Coot*[82] of selected individual sidechains and the PsbO loop region and placement of additional water molecules, the model was refined for several additional cycles with occupancy refinement enabled, then as before without automatic water-placement, and then as before with hydrogen atoms. NHQ flips and automatic linking were disabled throughout. A final "combined" dataset model was obtained with $R_{work}/R_{free}$ of 17.2%/21.7%.

In the final steps of refinement, Phenix-Auto-water-placement was used to model the waters. The water positions were manually inspected in COOT. The waters positioned within the discussed channels were moved to a different chain and renamed OOO. A bonding restraint CIF dictionary for OOO, identical to that for HOH, was supplied to Phenix.

With reset B-factors to 30 and removed waters, the above model was subsequently refined against the illuminated datasets with the lattermost refinement settings and different OEC bonding restraints. Using the 1.89 Å-model improved some important electron density features. These features were visible initially at a lower sigma level (<3), but after improving the model they mostly were present at higher sigma level (>4); i.e., W150, W151. OEC bonding restraints for the 0 F dataset prevented large deviations from the high-resolution dark state OEC structure reported by Suga *et al.* (PDB ID: 4UB6)[42]. Bonding restraints for the other datasets loosely restrained the models to metal-metal distances matching spectroscopic data and metal-oxygen distances matching the most likely proposed

models[83–87]. A number of ordered water positions were excluded from subsequent automatic water-placement rounds by renaming the residue names to OOO and the waters coordinating the OEC were incorporated into the OEC restraint CIF file directly. Density for an additional oxygen present in the S₃ state, Ox as reported previously[12,71], was visible in the 2F(150 μs) and later time point data and the Ox atom was included in the model and in the OEC CIF restraints in the final refinement for these four states. After 12–15 of cycles of refinement in this manner, individual illuminated states at various resolutions were obtained ranging in $R_{work}/R_{free}$ from 17.95%/22.70% to 18.48%/23.92% (Supplementary Table 2).

As described before[71] to best approximate the contributions of dimers that did not advance to the next S-state due to illumination misses, for the 2F(150 μs), 2F(250 μs), 2F(400 μs), and 2F datasets, the models were split into A and B alternate conformers in regions of chains A/a, C/c and D/d surrounding (and including) the OEC. The main conformer was set at 0.3 (2 F (150 μs)), 0.5 (2 F (250 μs)), 0.7 (2F (400 μs)), and 0.75 (2F) occupancy and the minor conformer was set to give a total occupancy of 1. The major conformer was allowed to refine as usual, while the minor conformer was fixed during refinement and set to match the S₂ state obtained from the refined 1 F structure.

**Estimated positional precision**. To estimate the positional precision of the OEC atoms, we used *END/RAPID* to perturb the structure factors, in an approach similar to one we previously employed[71,88]. Depending on the resolution and the fraction of S₃ present, metal-metal distances at the OEC had standard deviations between 0.07 and 0.27 Å across these trials, while distances between OEC metals and coordinating ligands were found to have standard deviations between 0.09 and 0.25 Å (for details see[71]).

**Modeling of waters**. To ensure the reliability of the modeling of waters in the channels, individual polder omit maps were generated, using *phenix.polder*[89], and the peak height was calculated, using *ccp4.peakmax*[90,91]. Peaks above 5 σ were considered to indicate a water molecule in the final models. While peaks between 3 and 5 σ, were only considered if they are present in one of the S states, at least, in the same vicinity, with a peak higher than 5 σ. Waters with peaks below 3 σ were rejected, i.e., W39 in some datasets (Supplementary Table 3).

**B-factors of the waters**. To investigate the B-factors of waters in the 1.89 Å-model, the occupancies were set to 1, then the waters were refined, including B-factor refinement against the data and results for both monomers were evaluated. To investigate the influence of occupancy on the refinement, a parallel refinement in which both occupancy and B-factors were allowed to refine was performed. Both strategies showed similar trends (Data not shown).

Similar strategies were used to track the changes in the water B-factors for the different time-points models. Since we are comparing the B factors from different models at different resolutions, it was necessary to standardize them. First, the water B-factors were extracted from each model. Then, the normalized B-factors (norm B) of each model were calculated by applying a Z-score (Eq. 2) as described by Carugo et al.[41],

$$norm\, B = \frac{B - B_{avg}}{B_{std}} \quad (2)$$

$$with\, B_{avg} = \frac{\sum B}{n} \quad (3)$$

$$and\, B_{std} = \sqrt{\frac{\sum (B - B_{avg})^2}{n - 1}} \quad (4)$$

**FoFc difference omit density**. To estimate the peak-height of water omit densities, changes in the electron density at the water (W$_i$) position were obtained from the omit maps of W$_i$ using the *FFT* program from the *CCP4* package[90,91].

**Water numbering convention**. We utilized a water numbering scheme for all waters in the vicinity of the OEC that is consistent with the numbering used in refs. 2,9. Waters are numbered with increasing numbers indicative of their distance to the OEC along the channels starting with W19. As the PDB does not allow to retain identifiers for waters the numbering in each deposited coordinate file is different. Hence, we are providing a table (Supplementary Table 3) that correlates the numbering used in this work with the numbering in each of the deposited coordinate files.

**Reporting Summary**. Further information on research design is available in the Nature Research Reporting Summary linked to this article.

## Data availability

The atomic coordinates and structure factors have been deposited in the Protein Data Bank, www.pdb.org (PDB ID code 7RF1 for the combined data, 7RF2 for the 0F, 7RF3 for the 1F, 7RF4 for the 2F(50 μs), 7RF5 for the 2F(150 μs), 7RF6 for the 2F(250 μs), 7RF7

for the 2F(400 μs), and 7RF8 for the 2F(200 ms) data). Already published protein structures used in this study can be found in the Protein Data Bank under accession codes 3JCU, 3WU2, 4UB6, 5TIS, and 6JLJ. Source data are provided with this paper.

## Code availability

The open source programs *dials.stills_process*, the *cctbx.xfel* GUI and *cxi.merge* are distributed with *DIALS* packages available at http://dials.github.io, with further documentation available at http://cci.lbl.gov/xfel.

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

## Acknowledgements

This work was supported by the Director, Office of Science, Office of Basic Energy Sciences (OBES), Division of Chemical Sciences, Geosciences, and Biosciences (CSGB) of the Department of Energy (DOE) (J.Y., V.K.Y., J.K.) for X-ray spectroscopy and crystallography data collection and analysis, and methods development for photosynthetic systems, by the National Institutes of Health (NIH) Grants GM055302 (V.K.Y.) for PS II biochemistry, GM110501 (J.Y.) and GM126289 (J.K.) for instrumentation development for XFEL experiments, GM117126 (N.K.S.) for development of computational protocols for XFEL data. N.K.S acknowledges support from the Exascale Computing Project (grant 17-SC20-SC), a collaborative effort of the DOE Office of Science and the National Nuclear Security Administration. R.H. acknowledges support by a Caroline von Humboldt Stipendium, Humboldt University Berlin. The NIH grants GM133081 (K.D.S.), GM124149 and GM124169 (J.M.H.), and Germany's Excellence Strategy (Project EXC 2008/1-390540038 (A.Z., H.D.) coordinated by T.U. Berlin and by the German Research Foundation (DFG) via the Collaborative Research Center SFB1078 (Humboldt Universität zu Berlin), TP A5 (A.Z., H.D., M.I., R.H.) and Vetenskapsrådet 2016-05183 (J.M.) and 2020-03809 (J.M.) are acknowledged for support. This research used resources of NERSC, a User Facility supported by the Office of Science, DOE, under Contract No. DE-AC02-05CH11231. XFEL data was collected at LCLS/SLAC, Stanford. Testing of crystals and various parts of the setup was carried out at synchrotron facilities that were provided by the ALS in Berkeley and SSRL in Stanford, funded by DOE OBES, and PL14 at BESSY, Berlin. The SSRL Structural Molecular Biology Program is supported by the DOE OBER, and by the NIH (P41GM103393). The Rayonix detector used at LCLS was supported by the NIH grant S10 OD023453. Use of the LCLS and SSRL, SLAC National Accelerator Laboratory, is supported by the U.S. DOE, Office of Science, OBES under Contract No. DE-AC02-76SF00515. We thank the support staff at LCLS/SLAC, SSRL (BL 6-2, 7-3) and ALS (BL 5.01, 5.0.2, 8.2.1, 8.3.1).

## Author contributions

R.A.M., U.B., N.K.S., J.M., A.Z., J.K., V.K.Y., and J.Y. designed the experiment; R.H., M.I., R.C., A.Z., and J.K. prepared samples; S.C., R.A.-M., and A. Batyuk operated the MFX instrument; I.-S.K., S.G., C.C.P., F.D.F., and J.K. developed, tested and ran the sample delivery system; M.H.C., R.C., C.d.L., P.C., J.M., and J.Y. characterized sample activity; R.H., M.I., A.B., R.C., L.L., I.-S.K., M.H.C., S.G., C.d.L., P.C., C.C.P., I.D.Y., S.C., F.D.F., R.A.-M., A. Batyuk, K.D.S., A.S.B., R.B., D.M., U.B., N.K.S., J.M., A.Z., J.K., V.K.Y., J.Y.

performed the XFEL experiment; A.B., A.S.B., R.B., D.M., J.M.H., N.W.M., P.D.A., and N.K.S. developed new software for data processing; A.B., L.L., I.D.Y., K.D.S., A.S.B., R.B., D.M., N.K.S processed XFEL data; R.H., M.I., A.B., P.S.S., L.L., M.D., I.B., K.D.S., H.D., A.Z., J.K., J.Y. analysed data. R.H., M.I., A.B., P.S.S., J.K., V.K.Y., and J.Y. wrote the manuscript with input from all authors.

## Competing interests

The authors declare no competing interests.
