## [Peer Review File · Nature Communications]

Structural Dynamics in the Water and Proton Channels of Photosystem II During the S2 to S3 TransitionREVIEWER COMMENTS

Reviewer #1 (Remarks to the Author):

This paper describes the possible roles of water channels that are linked with Mn₄CaO₅ in PSII based on the new high-resolution structure. Identification of proton transfer pathways and water intake pathways, both of which should originate from two substrate water molecules, is probably the most important topic in this field as it will ultimately clarify the water oxidation mechanism. Among structure-based studies, the main focus on water molecules is a novel viewpoint. Their proposal, the O1 channel serving as a water intake pathway and the Cl1 channel serving as a proton transfer pathway, is interesting, although they need to remove some speculation and controversy. The following points must be considered in the manuscript before recommendation.

Major points

1. In Figure 2, B-factor values for water molecules are shown in different colors. The authors may want to propose that water molecules are mobile specifically in the O1 channel. However, the current way is not a fair comparison, as the bar indicates that water molecules with the B factor values of ca. 30 Å² are already orange-colored while those with the slightly lower values of 27-28 Å² are white-colored. The authors should add B-factor values to water molecules in Figure (maybe better to make a new figure). In Figure, they don't have to show the entire channels but can show for example the regions within 15 Å from OEC (not only O1 but also O4 and Cl1). The reviewer suspects that the actual B-factor values of the orange balls in the O1 channel and the white balls in other channels might not differ so much.

2. The reviewer agrees that a tightly H-bond water network is unlikely to serve as a water intake channel. However, it is not always true that water molecules with large B-factor values form a water intake channel: they may simply be mobile, not more than that. The authors should reconsider the current strong statement of the O1 channel serving predominantly as a water intake channel.

3. A controversy can also be seen the locations of the Cl1 proton transfer and O1 water intake channels. While the Cl1 channel is probably a proton transfer pathway from accumulated evidence, the function of the O1 channel is unidentified. If the O1 channel is indeed a water intake pathway (toward Mn1), how can the authors reason the spacial gap between the starting point of the Cl1 channel and the endpoint of the O1 channel? How can the substrate water molecule delivered along the O1 channel release the proton to the disconnected Cl1 channel? If not, the released proton from the delivered water molecule must be transferred back again along the O1 channel toward the bulk surface. They should state these points clearly. As far as the authors insist on the proposed role of the O1 channel, they must explain the non-scientific "magic" based on fundamental chemistry. If they cannot, refrain from insisting on that. The reviewer considers that the current work is still interesting irrespective of the speculation of the role of the O1 channel.

4. Do the authors consider that the single substrate water molecule, which may be incorporated into the Mn1 site, comes exactly from the bulk surface to OEC in the S₂/S₃ transition (over the distance of more than 10 Å)? It seems that the authors' proposal is always based on such a misconception throughout the manuscript. See for example Figure 7D (long blue curved arrow). Indeed, a long journey is not required for water intake action. That is, no drastic displacement of water molecules along the channel is required.

5. If the substrate water molecule is delivered along the O1 channel to the Mn1 site, where is the deprotonation site? After incorporation, O6 is deprotonated. See the following paper and discuss: Mandal, M. et al. *J. Phys. Chem. Lett.* 11 (2020) 10262. The reported PSII structures do not provide any appropriate deprotonation site for the substrate water molecule that may come from the O1 channel. As the authors' work is based on the protein structure, they should not escape from the serious discussion of the deprotonation site based on their own structure!

6. Page 4, line 94. The absence of PsbV and PsbU in higher plants implies that the proposed role of the O1 channel may not be true in higher plants. At least, they should show the existence of the O1 channel toward the bulk surface together with PsbP and PsbQ in a structure-based figure.

7. Page 12, line 333. "the changes of the interaction between the redox-active YZ and H190, due to the oxidation of YZ" The distance is not short (2.5 Å for a low barrier hydrogen bond: Saito, K. et al. *Biochemistry* 50 (2011) 9836) in that structure?? Please provide the distance and discuss the

chemistry behind that. One can discuss a relevant chemical species of the tyrosine histidine pair based on the Biochemistry paper.

Minor points

8. Page 5, line 126. Possible involvement of the closed-cubane during the S2/S3 transition should be discussed. Cite Boussac, A. (2019) *Biochim. Biophys. Acta* 1860, 508.
9. Page 6. "higher B-factor" "high B-factor". "high B-factor values only near the bulk at the lumen side" etc. Be more specific, provide always the values. It may also be better to add a list as a supporting Table in SI.
10. The mobility of water molecules in the O1 channel is already pointed out in ref. 28, which should be discussed in the corresponding sentences.
11. Page 7, line 184-. The following paper should also be cited: Nakamura et al. (2014) *Biochemistry* 53, 3131.
12. Page 8, lines 208-210. "Ca" is not clearly shown in Figure 3.
13. Page 15, line 412. "PCET". Spell out as it appears for the first time.

Reviewer #2 (Remarks to the Author):

The manuscript by Hussein and coworkers addresses the nature of water channels in photosystem II. The quality of the structures has now improved to the point where identification is possible for the channels that funnel water molecules and protons to and from the Mn-center that serves as the active site for water oxidation. This topic is an active area of research and the outcomes of this paper should be of interest to a broad community of scientists interested in not only photosynthesis but also how metalloproteins facilitate complex reactions. In general, the paper is well written, the figures are well presented, and the discussion points are clear. The data analysis as described was challenging but it appears to be reliable and well performed. The discussion of the channels is very complete and reads nicely. Therefore, I recommend publication after the paper addresses information that I believe should be included as detailed below.

I find the description of the diffraction data analysis to be lacking. The authors state "In the present study, we focus on the question of mobility of the waters surrounding the OEC. 112 We do that by combing the large data set we have previously acquired throughout the Kok cycle 113 to obtain a high-resolution structure at 1.89 Å." I would like to have a more complete explanation provided in the supplementary. What criterion was used as the individual data sets were combined? Were the reflections from the individual data sets averaged to create the combined data set? What was the Rmerge? How were the Rfree reflection identified, what percentage was used, and did the authors bias this set to reflect those assigned as Rfree in the individual data sets. Finally, how could the combined data set have a resolution limit of 1.89Å when none of the individual sets extend to 2Å? Where do the last shell reflection come from in the combined data set?

The authors explain how the water molecules were identified, but missing is a discussion of the reliability of the water placements in the final model? In addition to general statements, the authors should include with Table S3, measures of the quality of the water molecules in the different channels, for instance the Fsigma and B values.

The authors need to identify the special aspects of this paper in the Abstract. The interpretation of the structural information depends heavily upon previous structural and spectroscopic studies. The data sets were previously published, and the channels were already identified. The abstract needs a clear statement about the novel aspects of this paper. The current statement, "The D1-E65 sidechain... as a proton gate" is weak.

Reviewer #3 (Remarks to the Author):

The authors further develop their analysis of structural changes of the photosystem II by serial x-ray crystallography using a x-ray free electron laser. However, there is little redundancy here since the manuscript describes an important new development providing considerably higher time and structural resolution. This is allowing, for the first time, the observation of important structural changes, including the surrounding water molecules, which are accompanying the S2 to S3 transition of the water oxidation reaction. The nature of this photochemical intermediate has been the subject of considerable experimental and theoretical analysis and discussion most notably because it appears to be the step where the substrate water is inserted and poisoning it for the final photooxidation step yield molecular oxygen. With the new improved resolution the authors are able to provide better confirmation of the pathway of substrate water insertion. Perhaps more impressively, they are able to visualize what is likely to be a crucial structural rearrangement of amino acid side chains and water molecules that facilitates proton ejection in a way that may minimize the back reaction. This is a big step forward and the text is already clearly and succinctly written, the figures nicely made to illustrate this complicated model from the large dataset such that I have only minor suggestions:

1. While the supplementary materials contain a nicely detailed explanation of the data processing that allowed for refinement of the structure and the development of structural models for waters in the channels within the protein. This appears to be quite innovative and it would be useful to better point this out from the main text and also to more clearly summarize the technical approach in the supplemental section before describing the details. For example it is not clear to the non-expert, whether having the large dataset was utilized allowed not accessible two single crystal methods.
2. In addition the authors do a nice job of comparing the room temperature structures with those obtained at cryogenic temperatures. I'm not sure if I agree with the statement that if the water is highly mobile, that it would preclude proton transfer via the H-bond network between waters and thus force potential transfer through asparagine 298. Obviously, this is an important mechanistic point and the authors at least need to present the argument better to be more convincing.
3. As nice as they are, it may be possible to reduce the number of figures with some rearrangement. Likewise, the main text is already fairly succinct, but it should also be possible to condense further.

Point-by-Point response to reviewer comments for Hussein et al.

We thank the reviewers for the constructive comments they have provided. We have answered all the questions in detail, and a point by point response to the comments and the changes made in the manuscript are described below. We are listing below the reviewer's comments in black with our responses below them in blue.

Reviewer #1

This paper describes the possible roles of water channels that are linked with Mn₄CaO₅ in PSII based on the new high-resolution structure. Identification of proton transfer pathways and water intake pathways, both of which should originate from two substrate water molecules, is probably the most important topic in this field as it will ultimately clarify the water oxidation mechanism. Among structure-based studies, the main focus on water molecules is a novel viewpoint. Their proposal, the O1 channel serving as a water intake pathway and the C11 channel serving as a proton transfer pathway, is interesting, although they need to remove some speculation and controversy. The following points must be considered in the manuscript before recommendation.

We thank the reviewer for the comments, and we have addressed each of the points below and have described the changes we made in the revised manuscript.

1. In Figure 2, B-factor values for water molecules are shown in different colors. The authors may want to propose that water molecules are mobile specifically in the O1 channel. However, the current way is not a fair comparison, as the bar indicates that water molecules with the B factor values of ca. 30 Å² are already orange-colored while those with the slightly lower values of 27-28 Å² are white-colored. The authors should add B-factor values to water molecules in Figure (maybe better to make a new figure). In Figure, they don't have to show the entire channels but can show for example the regions within 15 Å from OEC (not only O1 but also O4 and C11). The reviewer suspects that the actual B-factor values of the orange balls in the O1 channel and the white balls in other channels might not differ so much.

We thank the reviewer for pointing out the choice of the color gradient. We have updated Fig. 2A by adding a finer color gradient. Also, as suggested by the reviewer, we included the B-factor values in Supplementary Fig. 2. We note that the basic message remains the same.

2. The reviewer agrees that a tightly H-bond water network is unlikely to serve as a water intake channel. However, it is not always true that water molecules with large B-factor values form a water intake channel: they may simply be mobile, not more than that. The authors should reconsider the current strong statement of the O1 channel serving predominantly as a water intake channel.

The authors agree with the reviewer's statement that the higher water mobility derived from the larger B-factor values is not a direct evidence of the O1 channel being the water intake channel. Rather, it is likely be one of the important factors for the channel to serve as a water intake pathway with a functional purpose. The higher B factor is one of several pieces that add up to our conclusion of the O1 channel as the water intake pathway, that includes the Fo-Fc peaks (page 6 bottom, lines 163-168, and page 11 bottom, lines 313-317), the displacement of the water molecules observed at different time points (Supplementary Fig. 5), and the fact that the presence of additives in the O1 channel affects mainly S₂ to S₃ and S₃ to S₀ transitions (page 12, 2nd paragraph, lines 324-331). Also, the appearance of a strong positive peak appears within the O1 channel, directly after the Ox insertion (page 12, 3rd paragraph, lines 332-334).

3. A controversy can also be seen the locations of the C11 proton transfer and O1 water intake channels. While the C11 channel is probably a proton transfer pathway from accumulated evidence, the function of the O1 channel is unidentified. If the O1 channel is indeed a water intake pathway (toward Mn1), how can the authors reason the spacial gap between the starting point of the C11 channel and the endpoint of the O1 channel? How can the substrate water molecule delivered along the O1 channel release the proton to the disconnected C11 channel? If not, the released proton from the delivered water molecule must be transferred back again along the O1 channel toward the bulk surface. They should state these points clearly. As far as the authors insist on the proposed role of the O1 channel, they must explain the non-scientific “magic” based on fundamental chemistry. If they cannot, refrain from insisting on that. The reviewer considers that the current work is still interesting irrespective of the speculation of the role of the O1 channel.

In our opinion it is most likely that substrate inlet to the OEC is a stepwise process, likely involving W3. We hypothesize that W3 is replaced by W4, and that is replaced by a water from the O1 channel (Fig. 7). In this case, there is no spatial gap between the C11 channel and the O1 channel, as these channels are interconnected by several water molecules notably W24 and W25 that are close to W2 and W3. In the revised manuscript, we made it clear that the deprotonation only happens while the water bound to Ca (or Mn) moves into the Ox site (page 16 bottom, line 462- 464). For the case of W3 insertion, Ugur et al.¹ as well as Siegbahn² have suggested on how the proton can be transferred to the C11 channel for proton release to the bulk. We also point out the extensive work based on FTIR by for example Rick Debus,³ demonstrating the extended H-bonding network leading from D61 along the Mn₄CaO₅ cluster to W3.

4. Do the authors consider that the single substrate water molecule, which may be incorporated into the Mn1 site, comes exactly from the bulk surface to OEC in the S2/S3 transition (over the distance of more than 10 Å)? It seems that the authors’ proposal is always based on such a misconception throughout the manuscript. See for example Figure 7D (long blue curved arrow). Indeed, a long journey is not required for water intake action. That is, no drastic displacement of water molecules along the channel is required.

Obviously, we do not suggest that the water is transported from the bulk to the Mn cluster in a single transition during a water insertion event. We rather propose, as explained above, that a specifically pre-bound water (W3 on Ca) is inserted, and that the position of this pre-bound water is then refilled by the next water in the channel. So, while the water that is inserting into the Ox position travels a short way, all other water molecules in the channel will also move in order to refill the original binding site of this water. On page 16 in the discussion of the O1 channel, we describe this stepwise insertion of water along the O1 channel. The arrow in Figure 7D suggests one possible stepwise pathway. To make this point clear, we added “stepwise water insertion pathway” into the figure legend to avoid any possible confusion for the readers.

5. If the substrate water molecule is delivered along the O1 channel to the Mn1 site, where is the deprotonation site? After incorporation, O6 is deprotonated. See the following paper and discuss: Mandal, M. et al. J. Phys. Chem. Lett. 11 (2020) 10262. The reported PSII structures do not provide any appropriate deprotonation site for the substrate water molecule that may come from the O1 channel. As the authors’ work is based on the protein structure, they should not escape from the serious discussion of the deprotonation site based on their own structure!

We think this is related to the 3rd comment. Understanding the deprotonation steps is indeed an interesting question. So based on our structural studies, we proposed that W1 is deprotonated at 150 μs after the 2nd flash releasing the proton to D1-61 since a low barrier proton release was suggested of W1 via D1-61⁴ (Fig.7). As explained above and in the revised manuscript, water is easier to deprotonate when bound to a

metal like Mn or Ca. Therefore, we think that water needs to be bound to either Ca or Mn prior to being inserted at the Ox site. One possibility that we presented in the current manuscript is that W3, which is bound to Ca, may be the one that becomes Ox, and during this process, a proton is released to the deprotonated W1. This proton can then be released to the bulk via the C11 channel via H-bonding networks previously described by others in the field based on FTIR studies and with computational studies⁵⁻⁸.

6. Page 4, line 94. The absence of PsbV and PsbU in higher plants implies that the proposed role of the O1 channel may not be true in higher plants. At least, they should show the existence of the O1 channel toward the bulk surface together with PsbP and PsbQ in a structure-based figure.

The authors agree with the reviewer on the importance of showing how the O1 channel is structurally conserved in cyanobacteria and plants. Using one of the available Cryo EM structures of plant PSII (PDB: 3JCU), using CAVER, we generated the channels that start near the Ca side and extend through the D1 subunit ending either between CP43 and PsbP subunits or D2, CP47, PsbQ and PsbP subunits (see newly added section “Analysis of O1 channel in cyanobacteria and higher plants”, SI page 1). Our results showed that the cyanobacterial O1 channel A and O1 channel B are structurally conserved as the channel that proceeds along CP43 and PsbP subunits and along the D2, CP47, PsbP subunits in plants (new Supplementary Fig. 1). The results also agree with the study of Sakashita et al.⁹ that shows the cyanobacterial O1-PsbU/V channel is structurally conserved as the O1-PsbP channel in plants.

7. Page 12, line 333. “the changes of the interaction between the redox-active YZ and H190, due to the oxidation of YZ” The distance is not short (2.5 Å for a low barrier hydrogen bond: Saito, K. et al. *Biochemistry* 50 (2011) 9836) in that structure?? Please provide the distance and discuss the chemistry behind that. One can discuss a relevant chemical species of the tyrosine histidine pair based on the *Biochemistry* paper.

We thank the reviewer for pointing this out, we modified the revised manuscript accordingly (middle of page 14). For the full detailed descriptions of the structural changes within this region, we referred to the study of Ibrahim et al., while highlighting the main change observed in the revised manuscript. Also, we referred to the nature of the Yz-His H-bond (line 387), citing the suggested reference.

Minor points

8. Page 5, line 126. Possible involvement of the closed-cubane during the S₂/S₃ transition should be discussed. Cite Boussac, A. (2019) *Biochim. Biophys. Acta* 1860, 508.

We did not include the role of closed-cubane during the S₂ to S₃ transition, mainly because we currently do not have any experimental evidence for the formation of the closed cubane structure from our room temperature crystal structures or other experimental results¹⁰⁻¹². Thus, including this structure becomes highly speculative. Therefore, in the revised manuscript, we cited some literature, and added a sentence that we do not see any indication for the closed cubane structure in our current results (page 1, middle, line 118-121).

9. Page 6. “higher B-factor” “high B-factor”. “high B-factor values only near the bulk at the lumen side” etc. Be more specific, provide always the values. It may also be better to add a list as a supporting Table in SI.

The question has been addressed in the answer to Q1. We included the B-factor values in Supplementary Fig. 2 as suggested by the reviewer. We also added the range of numbers where we describe ‘high(er)-B-factor’ (line 148).

10. The mobility of water molecules in the O1 channel is already pointed out in ref. 28, which should be discussed in the corresponding sentences.

The theoretical study of reference 28 mainly discusses possible water intake pathways starting from a non-physiological state in which all water molecules are removed, and the protein backbones are fixed. While we find the paper highly interesting and have cited the work four-times in the manuscript, we do not see how this approach can address the changes and mobilities of water molecules in the channels connected to the water insertion during the S_2 - S_3 transition.

11. Page 7, line 184-. The following paper should also be cited: Nakamura et al. (2014) *Biochemistry* 53, 3131.

We thank the reviewer for his suggestion. In the revised manuscript, we cited this reference in the discussion section related to the Yz network (line 388).

12. Page 8, lines 208-210. “Ca” is not clearly shown in Figure 3.

In the revised manuscript, Figure 3 is updated, and the Mn_4CaO_5 cluster is colored according to atom so that the Ca atom is now visible as a green sphere.

13. Page 15, line 412. “PCET”. Spell out as it appears for the first time.

We thank the reviewer for pointing this out. In the revised manuscript this is now spelled out at the first usage, and the full wording is removed from line (461).

Reviewer #2

The manuscript by Hussein and coworkers addresses the nature of water channels in photosystem II. The quality of the structures has now improved to the point where identification is possible for the channels that funnel water molecules and protons to and from the Mn-center that serves as the active site for water oxidation. This topic is an active area of research, and the outcomes of this paper should be of interest to a broad community of scientists interested in not only photosynthesis but also how metalloproteins facilitate complex reactions. In general, the paper is well written, the figures are well presented, and the discussion points are clear. The data analysis as described was challenging but it appears to be reliable and well performed. The discussion of the channels is very complete and reads nicely. Therefore, I recommend publication after the paper addresses information that I believe should be included as detailed below.

We thank the reviewer for the very positive comments, and for recommending the manuscript for publication.

1-I find the description of the diffraction data analysis to be lacking. The authors state “In the present study, we focus on the question of mobility of the waters surrounding the OEC. We do that by combing the large data set we have previously acquired throughout the Kok cycle to obtain a high-resolution structure at 1.89 Å.” I would like to have a more complete explanation provided in the supplementary. What criterion was used as the individual data sets were combined? Were the reflections from the individual data sets averaged to create the combined data set? What was the Rmerge? How were the Rfree reflection identified, what percentage was used, and did the authors bias this set to reflect those assigned as Rfree in the individual data sets. Finally, how could the combined data set have a resolution limit of 1.89Å when none of the individual sets extend to 2Å? Where do the last shell reflection come from in the combined data set?

We appreciate the comment, which was also raised by Reviewer 3. In the revised manuscript, we added the detailed description of the diffraction data analysis to the SI as suggested (Section: X-ray diffraction data processing) (Page 3). We want to clarify that the collection of time resolved room temperature data from PSII is only possible using the serial crystallography approach we employed, relying on merging the data from many 1000s of individual crystals, each probed in a random orientation by an XFEL pulse. While a fraction of the individual images we collected showed diffraction out to high resolution better than 2.0 Å the available resolution for each of the individual time point data sets was limited by the number of images collected under each condition. In order to fully utilize this additional high-resolution information, we created a combined dataset by merging integrated intensities obtained from diffraction images collected under all different illumination conditions. We have included a more detailed description of how the combined dataset was obtained in the supplementary (Page 2, 3). The R_{merge} for the dataset is 11.7% (91.4% in the highest resolution shell, 1.922-1.890 Å). The R_{free} set was 0.89% (5584 reflections) of the total of 632624 reflections and was created by extending the resolution of the R_{free} set used for the refinement of the individual datasets. As a result, the R_{free} reflections for the combined dataset contain all the R_{free} reflections used for the individual datasets.

The higher quality of the combined dataset compared to the individual dataset is a consequence of how XFEL data is merged and the resolution cutoff is determined. The resolution for the combined dataset as well as the individual datasets are determined by a combination of criteria – (a) monotonic decrease of the $CC_{1/2}$ and (b) resolution where the multiplicity falls below 10. Each of the individual datasets had reflections that extended beyond 2 Å but the average multiplicity in those resolution shells for the individual datasets were below 10 and also the $CC_{1/2}$ was changing in a random manner. By combining the data measured under different illumination conditions, the statistics for the higher resolution shells were improved, which met the criterion used to determine the resolution cutoff, enabling us to merge the data out to a higher resolution.

2- The authors explain how the water molecules were identified, but missing is a discussion of the reliability of the water placements in the final model? In addition to general statements, the authors should include with Table S3, measures of the quality of the water molecules in the different channels, for instance the F_{sigma} and B values.

We thank the reviewer for the comment. In the revised manuscript, we included in the SI a full description that shows the main reliability criteria that we followed (Page 5, Modeling of Waters). Supplementary Tables 5 and 6 were added to show the peak height of the individual polder omit map calculated for each of the waters along the channels, and the value of the B-factor for the waters respectively.

3-The authors need to identify the special aspects of this paper in the Abstract. The interpretation of the structural information depends heavily upon previous structural and spectroscopic studies. The data sets were previously published, and the channels were already identified. The abstract needs a clear statement about the novel aspects of this paper. The current statement, “The D1-E65 sidechain... as a proton gate” is weak.

We thank the reviewer for raising this point. As suggested, we revised most of the abstract to address the novel aspects of this manuscript.

Reviewer #3

The authors further develop their analysis of structural changes of the photosystem II by serial x-ray crystallography using a x-ray free electron laser. However, there is little redundancy here since the manuscript describes an important new developments providing considerably higher time and structural resolution. This is allowing, for the first time, the observation of important structural changes, including

the surrounding water molecules, which are accompanying the S2 to S3 transition of the water oxidation reaction. The nature of this photochemical intermediate has been the subject of considerable experimental and theoretical analysis and discussion most notably because it appears to be the step where the substrate water is inserted and poisoning it for the final photooxidation step yield molecular oxygen. With the new improved resolution the authors are able to provide better confirmation of the pathway of substrate water insertion. Perhaps more impressively, they are able to visualize what is likely to be a crucial structural rearrangements of amino acid side chains and water molecules that facilitates proton ejection in a way that may minimizes the back reaction. This is a big step forward and the text is already clearly and succinctly written, the figures nicely made to illustrate this complicated model from the large dataset such that I have only minor suggestions:

We thank the reviewer for the very positive comments, and for recommending the manuscript for publication.

1. While the supplementary materials contain a nicely detailed explanation of the data processing that allowed for refinement of the structure and the development of structural models for waters in the channels within the protein. This appears to be quite innovative and it would be useful to better point this out from the main text and also to more clearly summarize the technical approach in the supplemental section before describing the details. For example, it is not clear to the non-expert, whether having the large dataset was utilized allowed not accessible two single crystal methods.

We appreciate the reviewer's comment and this suggestion, which was also pointed out by Reviewer #2. In the revised manuscript, we added a detailed description of the diffraction data analysis to the SI (Section: X-ray diffraction data processing) (Page 2, 3). We have included a more detailed description of how the combined dataset was obtained in the supplementary.

We want to clarify that the collection of time resolved room temperature data from PSII is only possible using the serial crystallography approach we employed, relying on merging the data from many 1000s of individual crystals, each probed in a random orientation by an XFEL pulse. In contrast, when using traditional single crystal methods all the different orientations necessary to obtain a complete dataset are collected from one single crystal. While this approach works for cryogenic conditions usually metalloenzymes, and especially PSII are too sensitive to radiation damage making collection of a complete data set on a single crystal at room temperature at a synchrotron impossible.

While a fraction of the individual images we collected showed diffraction out to high resolution better than 2.0 Å the available resolution for each of the individual time point data sets was limited by the number of images collected under each condition. In order to fully utilize this additional high-resolution information, we created a combined dataset, including diffraction images from all different illumination conditions. The combined data set created was primarily used to identify the water positions that were too weak for modeling in individual timepoints as well as to assess the mobility of the waters in the different channels. This is thus a novel aspect of this processing method.

2. In addition the authors do a nice job of comparing the room temperature structures with those obtained at cryogenic the temperatures. I'm not sure if I agree with the statement that if the water is highly mobile, that it would preclude proton transfer via the H-bond network between waters and thus force potential transfer through asparagine 298. Obviously, this is an important mechanistic point and the authors at least need to present the argument better to be more convincing.

We thank the reviewer for pointing this out, and as suggested we revised the manuscript to present our argument better as follows. We note that if this channel was a viable route for proton transfer to the bulk, the proton transfer would need to happen between waters W57 and W58 via N322. Our structural analysis showed that the distance between W57 and W58 is about 6.4 Å in the S₂ state, and in all the timepoints during S₂ to S₃ transition, we do not observe any substantial distance changes between these two waters and

their interaction with N322. Additionally, recent Quantum chemical calculations by Chrysinia et al.¹³ suggested tautomerization or amide rotations for the D1-N301-303-322 are needed to allow proton shifting through them. However, our RT structural data showed no conformational changes for the amide bond of D1-N(191, 301, 303, or 322) or along the Yz network during the S₂ to S₃ transition (Supplementary Fig. 12). This would serve as evidence that in all likelihood the Yz network is not a proton relay network. In addition to SI Fig. 12 we also included a more detailed discussion of the Yz network on page 14, middle, lines 392-398.

3. As nice as they are, it may be possible reduce the number of figures with some rearrangement. Likewise, the main text is already fairly succinct, but it should also be possible to condense further.

Each figure is quite condensed already and we could not come up with a way to further condense the figures, without making them difficult to follow. We think that to reach the general readership with this rather complicated scenario of waters and protons moving along channels, it may be necessary to include these figures. We note that the number of figures is within the limits prescribed by the journal guidelines.

1. Ugur, I., Rutherford, A.W., Kaila, V.R. Redox-coupled substrate water reorganization in the active site of photosystem II—the role of calcium in substrate water delivery. *Biochim. Biophys. Acta* **1857**, 740-748 (2016).
2. Siegbahn, P.E. The S₂ to S₃ transition for water oxidation in PSII (photosystem II), revisited. *Phys. Chem. Chem. Phys.* **20**, 22926-22931 (2018).
3. Kim, C.J., Debus, R.J. Evidence from FTIR difference spectroscopy that a substrate H₂O molecule for O₂ formation in photosystem II is provided by the Ca ion of the catalytic Mn₄CaO₅ cluster. *Biochemistry* **56**, 2558-2570 (2017).
4. Siegbahn, P.E. Mechanisms for proton release during water oxidation in the S₂ to S₃ and S₃ to S₄ transitions in photosystem II. *Phys. Chem. Chem. Phys.* **14**, 4849-4856 (2012).
5. Ishikita, H., Saenger, W., Loll, B., Biesiadka, J., Knapp, E.-W. Energetics of a possible proton exit pathway for water oxidation in photosystem II. *Biochemistry* **45**, 2063-2071 (2006).
6. Kuroda, H., *et al.* Proton transfer pathway from the oxygen-evolving complex in photosystem II substantiated by extensive mutagenesis. *Biochim. Biophys. Acta* **1862**, 148329 (2021).
7. Guerra, F., Siemers, M., Mielack, C., Bondar, A.N. Dynamics of Long-Distance Hydrogen-Bond Networks in Photosystem II. *J. Phys. Chem. B.* **122**, 4625-4641 (2018).
8. Okamoto, Y., Shimada, Y., Nagao, R., Noguchi, T. Proton and Water Transfer Pathways in the S₂→ S₃ Transition of the Water-Oxidizing Complex in Photosystem II: Time-Resolved Infrared Analysis of the Effects of D1-N298A Mutation and NO₃-Substitution. *J. Phys. Chem. B.* (2021).
9. Sakashita, N., Watanabe, H.C., Ikeda, T., Ishikita, H. Structurally conserved channels in cyanobacterial and plant photosystem II. *Photosyn. Res.* **133**, 75-85 (2017).
10. Ibrahim, M., *et al.* Untangling the sequence of events during the S₂ → S₃ transition in photosystem II and implications for the water oxidation mechanism. *Proc. Natl Acad. Sci. USA* **117**, 12624-12635 (2020).
11. Suga, M., *et al.* Light-induced structural changes and the site of O=O bond formation in PSII caught by XFEL. *Nature* **543**, 131-135 (2017).
12. Kern, J., *et al.* Structures of the intermediates of Kok's photosynthetic water oxidation clock. *Nature* **563**, 421-425 (2018).
13. Chrysinia, M., de Mendonça Silva, J.C., Zahariou, G., Pantazis, D.A., Ioannidis, N. Proton translocation via tautomerization of Asn298 during the S₂-S₃ state transition in the oxygen-evolving complex of photosystem II. *J. Phys. Chem. B* **123**, 3068-3078 (2019).

REVIEWER COMMENTS

Reviewer #1 (Remarks to the Author):

The authors have mostly addressed my concerns adequately. No further review is needed.

Reviewer #2 (Remarks to the Author):

The authors addressed all of the points raised previously in a comprehensive manner. I recommend publication.

Reviewer #3 (Remarks to the Author):

The authors have addressed the concerns that I have had. And importantly, the insightful criticisms of the other reviewers appears to be adequately addressed. The important issues of water and proton transport, especially regarding the diffusive nature of the transitions are better taken into account now.